# Label-free morpho-molecular phenotyping of living cancer cells by combined Raman spectroscopy and phase tomography
Arianna Bresci [1,2] ✉, Koseki J. Kobayashi-Kirschvink[1,3], Giulio Cerullo [2,4], Renzo Vanna [4], Peter T. C. So [1,5,6], Dario Polli [2,4] ✉ & Jeon Woong Kang [1] ✉

Accurate, rapid and non-invasive cancer cell phenotyping is a pressing concern across the life sciences, as standard immuno-chemical imaging and omics require extended sample manipulation. Here we combine Raman micro-spectroscopy and phase tomography to achieve label-free morpho-molecular profiling of human colon cancer cells, following the adenoma, carcinoma, and metastasis disease progression, in living and unperturbed conditions. We describe how to decode and interpret quantitative chemical and co-registered morphological cell traits from Raman fingerprint spectra and refractive index tomograms. Our multimodal imaging strategy rapidly distinguishes cancer phenotypes, limiting observations to a low number of pristine cells in culture. This synergistic dataset allows us to study independent or correlated information in spectral and tomographic maps, and how it benefits cell type inference. This method is a valuable asset in biomedical research, particularly when biological material is in short supply, and it holds the potential for non-invasive monitoring of cancer progression in living organisms.

Label-free live cell imaging techniques have revolutionized the field of cell biology by providing non-invasive tools to probe unperturbed and dynamic morpho-chemical traits of living cells[1]. The ability to study cellular processes without the need for any exogenous label not only preserves the integrity and native physiology of the cells but also eliminates potential artifacts associated with labeling[1]. Currently, fluorescent label-based microscopy has a leading role in the study of functional processes and morphological aspects of cells, but it comes with inevitable photo-bleaching, difficult signal reproducibility, some extent of photo-toxicity, and a limited amount of molecules to be targeted in parallel[2]. Several advantages are gained when samples do not need any manipulation and/or fixation: one can monitor time-dependent processes over an extended period, preserving the biological material unperturbed for further analysis or expansion, saving time and costs[3]. In recent years, significant technological advances in label-free imaging modalities have propelled the exploration of label-free morpho-molecular cell profiling at unprecedented levels of spatial and temporal detail[4].

Among these advanced label-free imaging methods, chemical imaging techniques enable the visualization and analysis of molecular constituents within living cells by leveraging the inherent chemical contrast given by biomolecules interacting with impinging light[5]. Chemical imaging of cells via spontaneous Raman scattering spectroscopy (RS) is acknowledged as one of the most informative techniques to describe cellular composition, organization, and function, by exploiting the Raman scattering effect[6,7]. It consists of the inelastic scattering of a small fraction of red-shifted photons ($\approx$1 out of $10^6$ photons) by molecules interacting with monochromatic laser light. This change in photon energy corresponds to the vibrational energy levels of the molecular bonds. In the 1980s, researchers began exploring the use of RS for biological studies[8,9]. Due to its small cross-section, spontaneous RS suffers from long exposure times, especially when dealing with a low density of scatterers. However, for detailed investigations of cell signatures in the highly informative fingerprint region of the Raman spectrum (Raman shift ($\Omega$) = 600 cm$^{-1}$–1800 cm$^{-1}$), spontaneous RS is the most widespread

[1]G. R. Harrison Spectroscopy Laboratory, Massachusetts Institute of Technology, Cambridge, MA 02139, USA. [2]Department of Physics, Politecnico di Milano, Milan 20133, Italy. [3]Klarman Cell Observatory, Broad Institute of MIT and Harvard, Cambridge, MA 02142, USA. [4]CNR-Institute for Photonics and Nanotechnologies (CNR-IFN), Milan 20133, Italy. [5]Department of Mechanical Engineering, Massachusetts Institute of Technology, Cambridge, MA 02139, USA. [6]Department of Biological Engineering, Massachusetts Institute of Technology, Cambridge, MA 02139, USA. ✉e-mail: abresci@mit.edu; dario.polli@polimi.it; jwkang76@mit.edu

method for an effective, non-invasive, accurate, and high signal-to-noise ratio chemical profiling[10].

Notably, the morphological arrangement of cells carries biomechanical information that contributes to describing their status and fate[11]. Research efforts have been put in place to develop solutions to reveal the shape, size, and structural properties of cells in a quantitative fashion without altering their natural state or introducing artifacts that may affect their behavior. Cutting-edge technology addressing this challenge sees quantitative phase imaging (QPI) at the forefront[12]. The development of QPI unfolded through a few milestones[13]: in the 1930s, the Dutch physicist F. Zernike invented phase-contrast microscopy, which allowed for the visualization of transparent samples by converting phase variations into intensity differences[14]. Interference microscopy came in the 1950s and expanded the possibilities of phase imaging[15]. By splitting a light beam into object and reference beams and recombining it after interacting with the sample, interference patterns were created, delivering quantitative information about phase variations. In the following decade, the invention of the laser led to the development of digital holography, which enabled the reconstruction of both the intensity and phase from a recorded interference pattern[16]. QPI kept evolving and now encompasses various techniques, producing either two-dimensional (2D) maps of the projected refractive index (RI) (holographic phase microscopy (HPM)[17] and diffraction phase microscopy (DPM)[18]) or more informative three-dimensional (3D) RI tomograms (tomographic phase microscopy (TPM)[19,20]). However, the major drawback of QPI techniques consists of a lack of chemical selectivity: by measuring the sample RI, it delivers an indirect measurement of structural and chemical changes that is difficult to interpret and compare with standard analytical methods.

These considerations pushed research towards the use of multimodal microscopy approaches to couple the benefits of label-free chemical and morphological imaging. In 2011, Kang et al. successfully coupled RS with HPM[21]. They showed how cell thickness correlates with hemoglobin distribution in healthy red blood cells (RBC) and with hemozoin distribution in malaria-infected RBC types. HPM merges digital holography and microscopy to measure the optical thickness and 2D RI variations within the sample. More recently, the same group compared co-registered HPM and RS information on a set of human cancer cell lines[22]: holographic maps showed high qualitative similarity with chemical images reconstructed from the Raman peak of proteins. In 2019, multimodal morpho-molecular imaging was also achieved by Pandey and coworkers proposing coupled DPM and RS to observe cell types in living conditions[23]. DPM, introduced in 2006 by G. Popescu, combines the single-shot nature of HPM with a common-path geometry, allowing for the measure of the optical phase with higher sensitivity[24].

A shared aspect of these label-free multimodal approaches and similar reports[21,25] is the use of 2D QPI maps, capable of delivering only an integrated projection of the RI along sample thickness, to extract the morphological counterpart of the cell profile. On the other hand, TPM achieves 3D morphological imaging[20,26], reconstructing a tomogram from multiple 2D phase images acquired at different illumination angles. 3D QPI unlocks additional quantitative data about cell morphology (e.g., cell volume, surface area, cell dry mass and density), and we consider that its combination with biomolecular fingerprints holds the potential of describing cell types in greater detail, boosting the accuracy of phenotype inference, even with limited population sampling. A comprehensive experimental proof of the potential of merging 3D QPI and RS information to profile cells for morpho-molecular phenotyping is highly sought in the field.

Here, we address the urge of developing a comprehensive, accurate, and non-perturbative cancer cells characterization tool by proving the superiority of co-registered RS and TPM in distinguishing quickly and quantitatively cancer cells having similar phenotypes (i.e., a non-trivial cell type discrimination task) (Fig. 1). We demonstrate that our approach can rapidly characterize and discriminate different human colon cancer cells, at progressing disease stages (i.e., HT29 colon adenoma, RKO colon carcinoma, and T84 lung metastasis of colon carcinoma), using a limited number of label-free living cells. We consider our results valuable for phenotyping tasks in biomedical research when in need of a rapid and non-perturbative method and in short supply of biological material.

## Results

### Quantitative morpho-chemical cell mapping via co-registered RS and TPM

RS and TPM maps were sequentially acquired on the same target cell, thus extracting co-registered morpho-chemical information (Fig. 2). For each human colon cancer cell line (i.e., HT29, RKO, and T84), five living cells are considered (Supplementary Fig. 1, Supplementary Fig. 2, and Supplementary Fig. 3), in order to reduce the materials and the measurement time while sampling biological variability[27,28] (Supplementary Fig. 4 and Supplementary Fig. 5).

Raman maps were obtained through a home-built confocal Raman microscope working at 785-nm continuous-wave laser excitation[29] (Methods, RS system). RS data are 3D XY-Ω hypercubes, where each pixel in the XY spatial plane is composed of a spectrum in the fingerprint

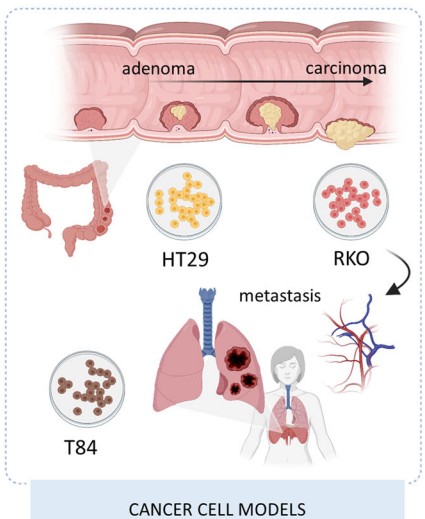
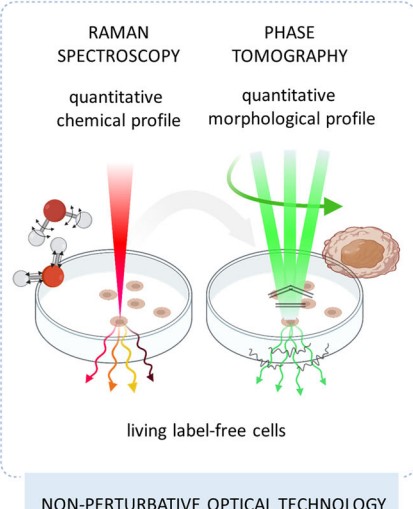
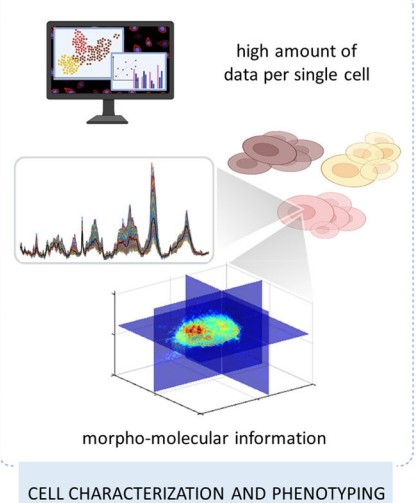

**Fig. 1 | Experimental design.** RS and TPM are merged on the same field of view to extract the morpho-chemical profile of cancer cells for quick cell characterization and phenotype discrimination. By exploiting the high level of quantitative detail that RS and TPM provide about the same single cell, we demonstrate a rapid and statistically significant discrimination of cancer cell types using cells in label-free, living, and unperturbed conditions. Created with BioRender.com.

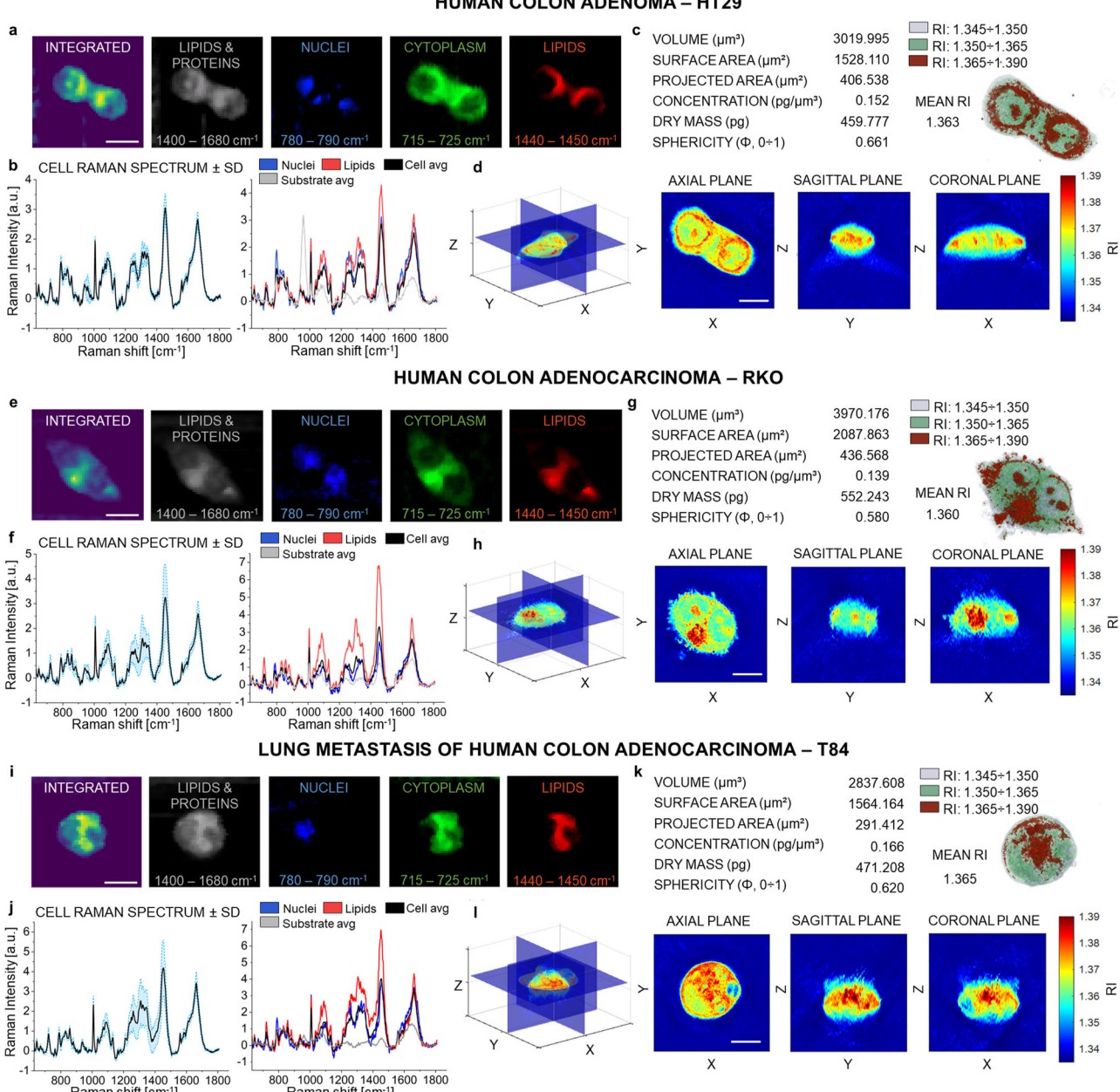

**Fig. 2 | Representative morpho-chemical information collected through RS and TPM in human colon cancer-derived cells.** Cell cycles were not synchronized through the work, so as to make results agnostic to sampled cell cycle phases (Supplementary Fig. 12 and Supplementary Fig. 13). **a, e, i** RS microscopy images of an HT29, RKO, and T84 cell, obtained integrating the hypercubes along the $\Omega$ dimension in the indicated spectral region. **b, f, j** On the left, cell average spectrum (in black) ± standard deviation (in blue), and single-point spectra of nuclei (in blue, $\Omega$ = 780–790 cm$^{-1}$) and lipids (in red, $\Omega$ = 1440–1450 cm$^{-1}$), compared to the cell mean (in black) and substrate mean (in grey). **c, g, k** Set of quantitative information extracted from the TPM RI maps of the cell. **d, h, l** Sections of the cell tomogram, along the axial, sagittal, and coronal planes. Scale bars: 10 µm in both RS and TPM images.

region of the Raman vibrational profile of biological matter ($\Omega$ = 600 cm$^{-1}$ to $\Omega$ = 1800 cm$^{-1}$) (Fig. 2b, f, j). In this work, 50 × 50 Raman spectra per each single-cell field of view (FOV) (i.e., 40 × 40 µm$^2$ with a pixel size of 800 × 800 nm$^2$) were collected with a 1.5-seconds pixel dwell time (PDT) to obtain sufficient signal-to-noise ratio for rigorous chemical interpretation down to single-pixel spectra (Fig. 2b, f, j). In fact, clear differences can be seen in single DNA and RNA pixels (Fig. 2b, f, j in blue), and lipid ones (Fig. 2b, f, j, in red). By sectioning along the $\Omega$ axis, we could retrieve false-color images depicting the spatial distribution of major subcellular components (Fig. 2a, e, i). These include: (i) generic organic cellular matter[30,31] in the range $\Omega$ = 1400–1680 cm$^{-1}$ (Fig. 2a, e, i, grey channel), with proteins signatures of Amide I and C = C stretching at 1656 cm$^{-1}$, phenylalanine and tyrosine at 1604 cm$^{-1}$, and CH vibrations of aliphatic side chains at 1448 cm$^{-1}$; (ii) lipid accumulations in the range $\Omega$ = 1440–1450 cm$^{-1}$, due to the bending of CH$_2$ bonds in fatty acids (Fig. 2a, e, i, red channel); (iii) nuclei in the $\Omega$ = 780-790 cm$^{-1}$ region[32,33], corresponding to pyrimidines ring breathing, including both nucleobases of DNA and RNA cytosine and uracyl at $\Omega$ = 788 cm$^{-1}$ (Fig. 2a, e, i, blue channel); (iv) cytoplasm[34] at $\Omega$ = 715–725 cm$^{-1}$, featuring Raman modes of CN$^+$(CH$_3$)$_3$ in lipids, choline group N$^+$(CH$_3$)$_3$ and phosphatidylcholine vibrations (Fig. 2a, e, i, green channel). RS maps distinguish the targeted cell from the substrate, even in presence of phosphate salts in the culture medium that generate background peaks at 970 cm$^{-1}$ (Fig. 2b, grey spectrum).

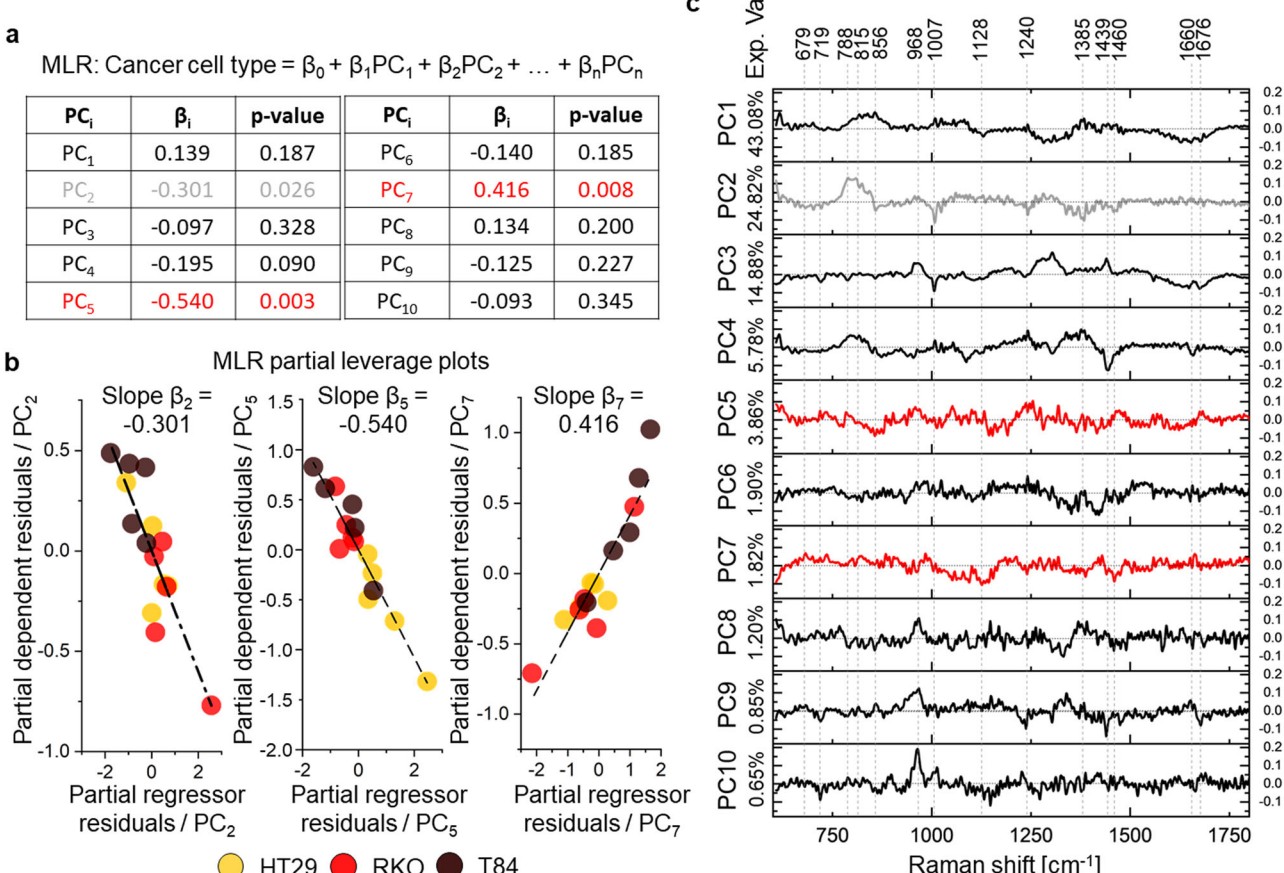

**Fig. 3 | PCA-MLR analysis results on the dataset of cell-averaged Raman spectra measured in living human colon cancer cell lines. a** MLR fitting equation and estimated model coefficients ($\beta_i$) with their statistical significance ($p$-values) by via a two-tailed Student $t$-test. In grey, low-significance ($p$-value < 0.05) $\beta$ coefficients. In red, high-significance ($p$-value < 0.01) $\beta$ coefficients. **b** Partial leverage plots of significant PCs in the MLR model. **c** PCA loadings with indication of relevant Raman peaks. Red PCs are associated to highly significant $\beta$ coefficients, the grey PC features lower significance. Sample size = 5 independent cells/phenotype.

On the other hand, TPM delivers cell tomograms, in which the RI of the specimen constitutes the value of each voxel through the sample volume[13]. We employed a holo-tomographic microscope (Methods, TPM system) working with a 532-nm laser excitation. TPM images consist of a 3D XYZ RI datacube. Once obtained the RI tomogram, we segmented cells from the aqueous substrate via proper thresholding ($RI_{threshold}$= 1.345) to estimate quantitative values of cell volume (V) [$\mu m^3$], surface area (S) [$\mu m^2$], projected area (i.e., footprint) (A) [$\mu m^2$], and sphericity ($\Phi$=0-1) (Fig. 2c, g, k). Thickness (T) [$\mu m$] was obtained via an ellipsoid approximation of the cellular shape, so that $T = 3/2 \cdot \frac{V}{A}$. The refractive properties of a cell ($n_{cell}$) exhibit a strong dependence on the total concentration of non-acqueous cellular content, mainly proteins[35]: $n_{cell}(x, y, z) = n_0 + \alpha C(x, y, z)$, where $n_0$ is the RI of the medium, $\alpha = 0.19$ ml/g is the well-established refraction increment [ml/g] of proteins[36], and C is the concentration of dry content [g/ml]. Hence, from the extracted values of C, we can estimated the dry mass density D [pg $\mu m^{-3}$] of the cell, as $D = \iiint C(x, y, z)dxdydz = \sum C(x, y, z)\Delta x\Delta y\Delta z$, where $\Delta x$, $\Delta y$ and $\Delta z$ represent the size of voxels above threshold (Fig. 2c, g, k). Having access to V and D, we computed the overall dry mass (DM) [pg] of the specimen as a product of these quantities. The tomogram slices along the axial (XY), sagittal (YZ) and coronal (XZ) plane of the cell reveal 3D subcellular structures (Fig. 2d, h, l).

In this work, the presented set of morpho-molecular descriptors was obtained from the same cell by systematically co-registering the two microscopy approaches. In the following, we show that such data carry enough information to rapidly discriminate similar cancer cell subtypes, easing the phenotyping task by measuring a limited amount of living cells in

their original form, avoiding any time-consuming manipulation, labeling and/or fixation.

**Extracting chemical information from RS for cell characterization**
Cell spectra obtained via RS are high-dimensionality data: for each XY point in the Raman image, the $\Omega$ dimension contains 870 wavenumbers. Dimensionality reduction of such a large dataset is needed to extract information relevant to cell type differentiation. We present an effective, simple, quick and human bias-free data-processing pipeline to derive chemical features that most represent the differences between the observed cancer cell types, leveraging Principal Component Analysis (PCA) and Multiple Linear Regression (MLR)[37,38] on cell-averaged Raman spectra.

PCA is a widely validated method in spectroscopy to reduce the dimension of hyperspectral Raman data, as it preserves the chemical interpretability of the new dataset[39]. It is not simply possible to consider the first principal components (PCs) as the most significant ones for phenotype discrimination[40], based on the fact that they explain the highest variance in the original data. To dissect how PCs relate to target phenotypes, we exploited MLR[41,42]: linear regression coefficients quantify the contribution of each PC in differentiating target variables (i.e., cell phenotype = $\beta_0 + \beta_1 \cdot PC_1 + \beta_2 \cdot PC_2 + \ldots + \beta_n \cdot PC_n$). We used as independent variables the first 10 PCs (Fig. 3a), which explain 98.9% of cumulative variance, then analyzed their MLR coefficients ($\beta_i$) statistics to identify a subset of significant PCs. This screening isolated three coefficients (Fig. 3a): $\beta_2 = 0.301$, $\beta_5 = -0.54$, and $\beta_7 = -0.416$, with $p = 0.026$, $p = 0.003$ and $p = 0.008$, respectively[43]: we selected $PC_2$, $PC_5$ and $PC_7$ (Fig. 3) as

**Article**

**Fig. 4 | PCA score plots analysis results on the dataset of cell-averaged Raman spectra measured in living human cancer lines.** By screening the first PCs through MLR, PCA score plots can be used to linearly separate human colon cancer cell types in the PCA space, based on their biochemistry. Sample size = 5 independent cells/phenotype. **a** The scatter matrix compares pair-wise the first 7 PCs, explaining 96% of the total dataset variance. Red dashed lines highlight pairs of the most significant PCs, namely, $PC_5$ and $PC_7$. Grey dashed lines identify the less significant PC, namely, $PC_2$. **b** The entire set of significant PCs is represented in the 3D PCA space.

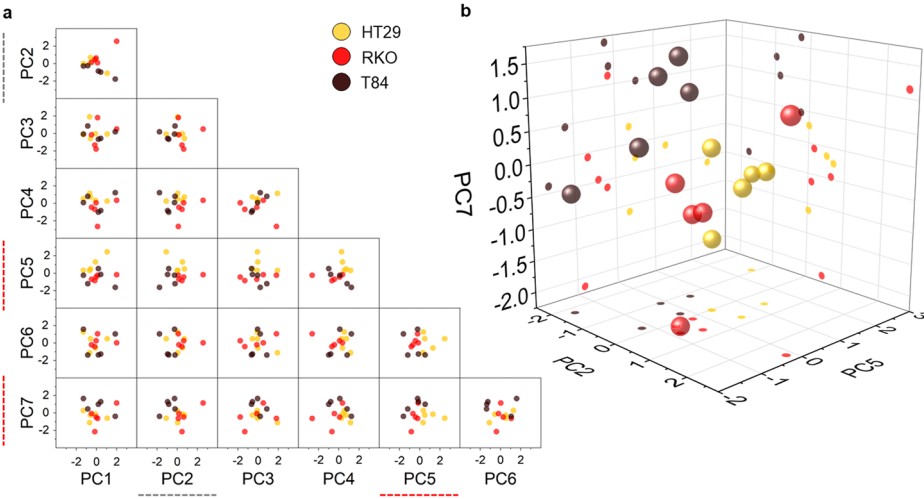

independent predictors of cell type. We analyzed MLR partial leverage plots, obtained as plots of the residuals of the dependent variable, omitting the selected regressor (i.e., the partial dependent residuals), against the residuals of the explanatory variable regressed on all the others independent variables (i.e., the partial regressor residuals), to corroborate this selection (Fig. 3b)[44]. Such plots identify outliers or high-leverage points that alone impact strongly on the regression line (Supplementary Fig. 6). One can appreciate how $PC_2$, $PC_5$ and $PC_7$ partial residuals are well fitted by the $\beta_i$ regression line (Fig. 3b), confirming the choice of these PCs as the most significant chemical information for cell type discrimination.

We biochemically interpreted the PCA-MLR output via PCA loadings. The i-th loading is composed by coefficients ranging between -1 and 1, that give the contribution of each Raman frequency toward the i-th PC. High absolute values indicate strong positive or negative correlation, while 0 indicates a weak influence[45] (Fig. 3c). $PC_1$ explains 43.08% of the total dataset variance (Fig. 3c), even though it is not a significant regressor in the MLR model, and it is mainly related to the presence of a broad quartz band ranging from $\Omega = 750$ cm$^{-1}$ to $\Omega = 900$ cm$^{-1}$ [46], which is the shared substrate of all samples, thus observing its spectral contribution is reasonable. Another correlation of $PC_1$ is with the peak of water ($\Omega = 1600$–$1700$ cm$^{-1}$) and general proteins (Amide III and Amide I vibrational energies at $\Omega = 1240$–$1280$ cm$^{-1}$ and $\Omega = 1628$–$1662$ cm$^{-1}$, respectively) (Fig. 3c)[47], constituents shared by the samples. On the other hand, $PC_2$, explaining 24.82% of the total variance (Fig. 3a), features a broad peak of quartz overwhelming DNA and RNA modes ($\Omega = 788$ cm$^{-1}$, $\Omega = 815$ cm$^{-1}$ and at $\Omega = 826$–$831$ cm$^{-1}$) (Fig. 3c), along with a relatively lower amount of proteins (phenylalanine at $\Omega = 1007$ cm$^{-1}$, Amide I at $\Omega = 1240$–$1280$ cm$^{-1}$, $CH_3$ in general proteins at $\Omega = 1385$ cm$^{-1}$ and $\Omega = 1460$ cm$^{-1}$), and fatty acids in lipids ($\Omega = 1440$ cm$^{-1}$ and $\Omega = 1460$ cm$^{-1}$). It suggests scarcity of cell matter in favor of background signal, which may be due to lower cell thickness (see Tab. 1). Its significance in distinguishing cell subtypes is the lowest in the pool of selected PCs. In fact, $PC_5$ and $PC_7$ alone are capable to separate cell types in the PCA space (Fig. 4). In particular, $\beta_5$ displays the highest statistical significance (i.e., $p = 0.003$) and the highest absolute value (i.e., $\beta_5 = -0.54$) (Fig. 3a). The $PC_5$ loading is negatively correlated with amino-acid vibrations of proline and hydroxyproline, as well as C-C vibration of collagen, at $\Omega = 856$ cm$^{-1}$, C-N vibrations in proteins and C-C stretching in lipids[34] at $\Omega = 1131$–$1152$ cm$^{-1}$. Conversely, at $\Omega = 1240$ cm$^{-1}$, $PC_5$ is positively related to nucleic acids, mostly RNA[34] (Fig. 3c) (Tab. 1). This aligns with $PC_2$ interpretation, attributing a relevant role in cancer cells discrimination to nucleic acids prevailing over proteins and lipids. $PC_7$ mainly shows a negative correlation with: (i) the C-N stretching vibration in proteins and to the C-C stretching modes of lipids and proteins[34] around $\Omega = 1030$–$1090$ cm$^{-1}$ and $\Omega = 1122$–$1160$ cm$^{-1}$, (ii) the Amide I band

($\Omega = 1240$–$1280$ cm$^{-1}$), as well as (iii) fatty acids in lipids ($\Omega = 1440$ cm$^{-1}$ and $\Omega = 1460$ cm$^{-1}$) (Tab. 1).

Overall, PCA-MLR attributes a leading role to DNA and RNA content in the diversification of colon cancer cell types, accompanied by relative variations of lipids and proteins amount. The effectiveness of the proposed PCs screening method is visually clear in scatter plots of PCs scores (Fig. 4): the significant $PC_5$ – $PC_7$ pair of regressors corresponds to the PCA plane with the highest spatial separation among phenotypes.

This PCA-MLR chemical interpretation is in line with reports using invasive characterization techniques such as destructive gene expression analysis on large cell populations. Lung metastatic colon cancer cells are distinguished from colon resident types by a downregulation of angiogenin and amphiregulin, lung-specific angiogenesis proteins[48]. This is in agreement with positive $PC_7$ scores in metastatic T84s, telling them apart from resident HT29s and RKOs (Fig. 4). Accordingly, previous research identifies nucleic acids variations distinguishing colon-resident cancer cell lines[49,50]: here, HT29s and RKOs are mainly separated along $PC_5$ (Fig. 4).

**Extracting morphological information from TPM for cell characterization.** Thanks to the 3D RI maps obtained through TPM, a set of quantitative descriptors of cell morphology can be derived through numerical calculation: the average RI, dry mass (DM) and density (D), sphericity ($\Phi$), volume (V), surface area (S), footprint (A), and thickness (T). It is worth noticing that the generation of RI tomograms and the subsequent extraction of data requires extended computational power, unlike the use of raw spectra in RS measurements. Indeed, first the field-retrieval algorithm is used to obtain 2D phase maps. Then, the diffraction tomography algorithm delivers 3D RI tomograms (see Methods, TPM system). This comes with the advantage of having ready-to-use morphological information, leading to a simplified data processing pipeline to discriminate cancer cell subtypes. TPM images are reduced to eight numerical observables, not requiring dimensionality reduction ahead of a feature importance study (Fig. 5).

On average, visual inspection via standard bright-field microscopy does not highlight any peculiar trait differentiating these colon cancer cells, except for a slightly more elongated shape shown by RKOs (Supplementary Fig. 7), and a slightly smaller and more roundish shape in HT29s. Conversely, TPM data reveal significantly different morphological features. HT29s display the highest morphological difference when compared to others (Tab. 1). Their average RI (Fig. 5a) is higher with respect to RKO and T84 types (RI$_{HT29}$ = 1.364 ± 6 × 10$^{-4}$, RI$_{RKO}$ = 1.359 ± 7 × 10$^{-4}$, and RI$_{T84}$ = 1.362 ± 0.001), and this translates into higher average D values (Fig. 5c) (D$_{HT29}$ = 0.159 ± 0.003 pg μm$^{-3}$, D$_{RKO}$ = 0.133 ± 0.004 pg μm$^{-3}$, and D$_{T84}$ = 0.148 ± 0.006 pg μm$^{-3}$). They also have higher

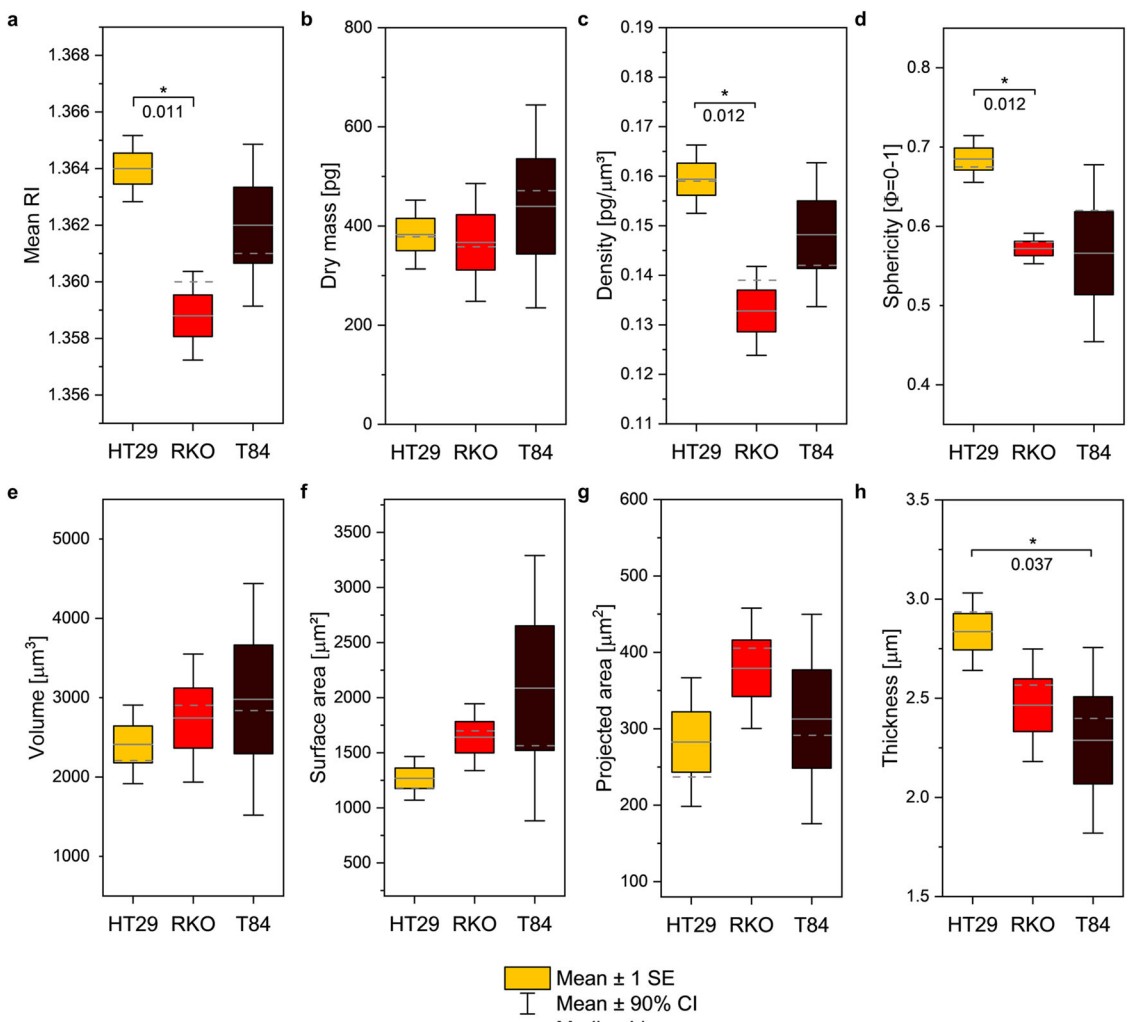

**Fig. 5 | TPM-derived quantitative morphological traits of HT29, RKO and T84 colon cancer cells and relative significant trends.** Box plots report the standard error of the mean (SE) of morphological indexes computed from TPM maps, whiskers represent the 90% confidence interval (CI), whereas grey solid and dashed lines represent the mean and median of the distribution, respectively. Sample size = 5 independent cells/phenotype. **a** mean cell RI, **b** cell DM [pg], **c** cell D [pg μm$^{-3}$], **d** cell Φ [0-1], **e** cell V [μm$^3$], **f** cell S [μm$^2$], **g** cell A [μm$^2$], **h** cell T [μm]. *P*-values are computed through the U-Mann Whitney non-parametric hypothesis test and are here shown for significant comparisons (\**p*-value < 0.05), otherwise consider significance not reached.

average sphericity (Φ) (Fig. 5d) ($\Phi_{HT29} = 0.685 \pm 0.014$, $\Phi_{RKO} = 0.572 \pm 0.009$, and $\Phi_{T84} = 0.566 \pm 0.052$). In line with this, HT29s are thicker (Fig. 5h) ($T_{HT29} = 2.835 \,\mu m \pm 0.092$, $T_{RKO} = 2.465 \,\mu m \pm 0.133$, and $T_{T84} = 2.289 \,\mu m \pm 0.220$). Figure 5 shows other morphological trends, despite not statistically significant. RKOs distinguish from both HT29s and T84s because of a lower D (Fig. 5c). T84s have lower Φ (Fig. 5d) and T (Fig. 5h), resulting in a larger S (Fig. 5f).

To better quantify the mutual relationship between these traits, we analyzed their Pearson Correlation Coefficients (PCCs), reflecting any linear correlation and their pairwise statistical significance (Fig. 6). Interestingly, although morphological traits do not exhibit highly different trends among cancer subtypes, their correlation distinguishes cell types more clearly.

Excluding trivial correlations, such as D and RI pairs and diagonal self-correlations, significant PCCs pairs (\*) vary across cell types. A significantly positive PCC characterizes dry mass (DM) and volume (V) in all phenotypes (HT29: PCC = 0.982, p = 0.003; RKO: PCC = 0.983, p = 0.003; T84: PCC = 0.987, p = 0.002): regardless of the subtype, an enlarged V is associated with a higher DM content. In line with this, also A and S pair with DM in a strongly positive PCC. Other PCCs exhibit values unique to some phenotypes. While T84s feature a negligible PCC relating D with V, S, and

A, we can observe a D reduction in HT29s as they increase their overall dimension, whereas RKOs tend to increase their D, in an opposite trend (Tab. 1). As for cell shape, a peculiar morphological trait of T84s is a clear negative correlation between Φ and S, a trend shared by HT29s but not present in RKOs (HT29: PCC = −0.498, p = non-significant (ns); RKO: PCC = 0.224, p = ns; T84: PCC = −0.973, p = 0.005). Similarly, cell V and A feature positive PCC values with Φ only in RKO cells; this entails a peculiar morphological feature: as the dimension increases in RKOs, they acquire a more spherical shape. Vice versa, as V, S, and A in HT29s and T84s increase, they lose their spherical appearance (Tab. 1). When comparing trends (Fig. 5) though, these phenotypes feature similar cell dimension, with RKOs having a lower sphericity. Indeed, one can tell apart RKOs as the less roundish type in the XY plane, appreciable also through bright-field inspection (Supplementary Fig. 7) but here statistically quantified via TPM. Also, T84s distinguish by their negative correlation between T and other dimensionality indexes (i.e., V, S, and A): the physical interpretation of this results is that, as they grow bigger in dimension, they tend to spread on the culture substrate decreasing their thickness, a behavior not shown in HT29s and RKOs.

TPM unlocks a highly detailed quantitative and comparative morphological description of different colon cancer cells. In line with our

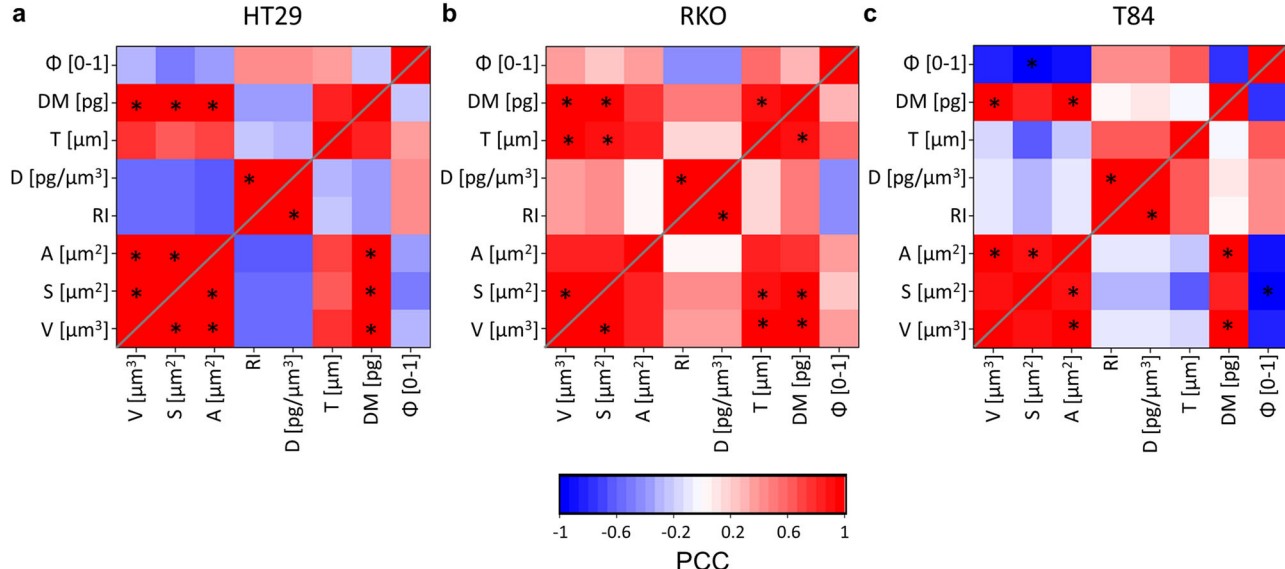

**Fig. 6 | Pearson correlation matrices of TPM-derived quantitative indexes.** The heatmaps report PCC values for morphological indexes pairs. In blue, negative correlation (PCC < 0). In red, positive correlation (PCC > 0). Resident colon cells, namely, HT29s **a** and RKOs **b** and metastatic T84s colon cancer cells **c** PCC heatmaps. Statistical significance (*$p$-value < 0.05) is computed via a two-tailed Student $t$-test and indicated for each significantly correlated pair. Sample size = 5 independent cells/phenotype.

**Table 1 | Quantitative morpho-molecular traits differentiating colon cancer-derived cell types**

| Technique | Cell trait | HT29 | RKO | T84 |
|---|---|---|---|---|
| RS (chemical) | DNA and RNA relative abundance over C-C and C-N in proteins and lipids (PC$_5$) | ✓ | × | × |
| RS (chemical) | Thin cell, relative scarcity of cellular organic matter (PC$_2$) | ✓ | ✓ | × |
| RS (chemical) | Relative scarcity of proteins and lipids, C-C and C-N, CH$_2$ and CH$_3$ (PC$_7$) | × | × | ✓ |
| TPM (morphological) | Relatively high D, T, and Φ | ✓ | × | × |
| TPM (morphological) | Cell D correlation with cell dimension | ✓ (−) | ✓ (+) | × |
| TPM (morphological) | Cell Φ correlation with cell dimension | ✓ (−) | ✓ (+) | ✓ (−) |

The table summarizes the main characteristics of colon cancer-derived cell types that set them apart on a morpho-chemical basis. ✓ and × indicate presence or absence of a cell trait, respectively. In case of correlation traits, (−) and (+) further specify negative or positive correlation, respectively.

observations, Zadka et al. previously showed how HT29s set apart from other colorectal cancer cells, not including the ones here used, by having higher RI and D values, similarly quantified here via TPM[51].

## Cell type discrimination via combined morpho-molecular profiling

In this work, besides showing how chemical and morphological information can be independently extracted and interpreted (Table 1) via RS and TPM to rapidly differentiate similar living cancer cells, we show that integrating this synergistic insight leads to superior cell phenotype inference.

To this aim, we quantitatively estimated phenotype inference through multinomial logistic regression (MLoR) mathematical modeling. This model, fed with independent chemical, morphological, and, ultimately, combined morpho-molecular data, can tell apart the colon cancer types (HT29s, RKOs, or T84s) by estimating the phenotype probability. To evaluate competing MLoR models, election methods are the Likelihood Ratio Test (LRT) or the Akaike's Information Criterion (AIC). The LRT probability can be used to estimate the significance of the inference. The AIC quantifies model accuracy weighted by its complexity: it is directly

proportional to the number of explanatory variables, so that a simpler model receives a lower penalty resulting in a lower AIC.

First, we test the entire set of significant RS and TPM information, namely, the most significant PCs and the most relevant morphological observables for RS- and TPM-based phenotype inference, respectively (Fig. 7a–c). When using the complete set of significant PCs, the chemical information is enough to explain the differences between cell types (Fig. 7a), with LRT $p$-value = $1.161 \times 10^{-5}$ and AIC = 16.188. Conversely, when using TPM information (Fig. 7b), the accuracy is significant (LRT $p$-value = 0.006), but AIC = 30.804 witnesses poor model fitting. Merging these two sets of explanatory variables (Fig. 7c) results in a more complex model, not improving any performance (LRT $p$-value = $9.865 \times 10^{-4}$, AIC = 28.011).

To further break down the impact of each piece of information for phenotype inference, we limit to only the most relevant RS and TPM explanatory variables, in simpler models that ease interpretation (Fig. 7d–f). The most significant RS and TPM information to distinguish cell types are PC$_5$ ($p$ = 0.003) (Fig. 3) and D, respectively. When training the MLoR model with PC$_5$, we observe a drop in performance (LRT $p$-value = 0.002, AIC = 28.416, and inference accuracy 66.7%) with respect of the full RS-based

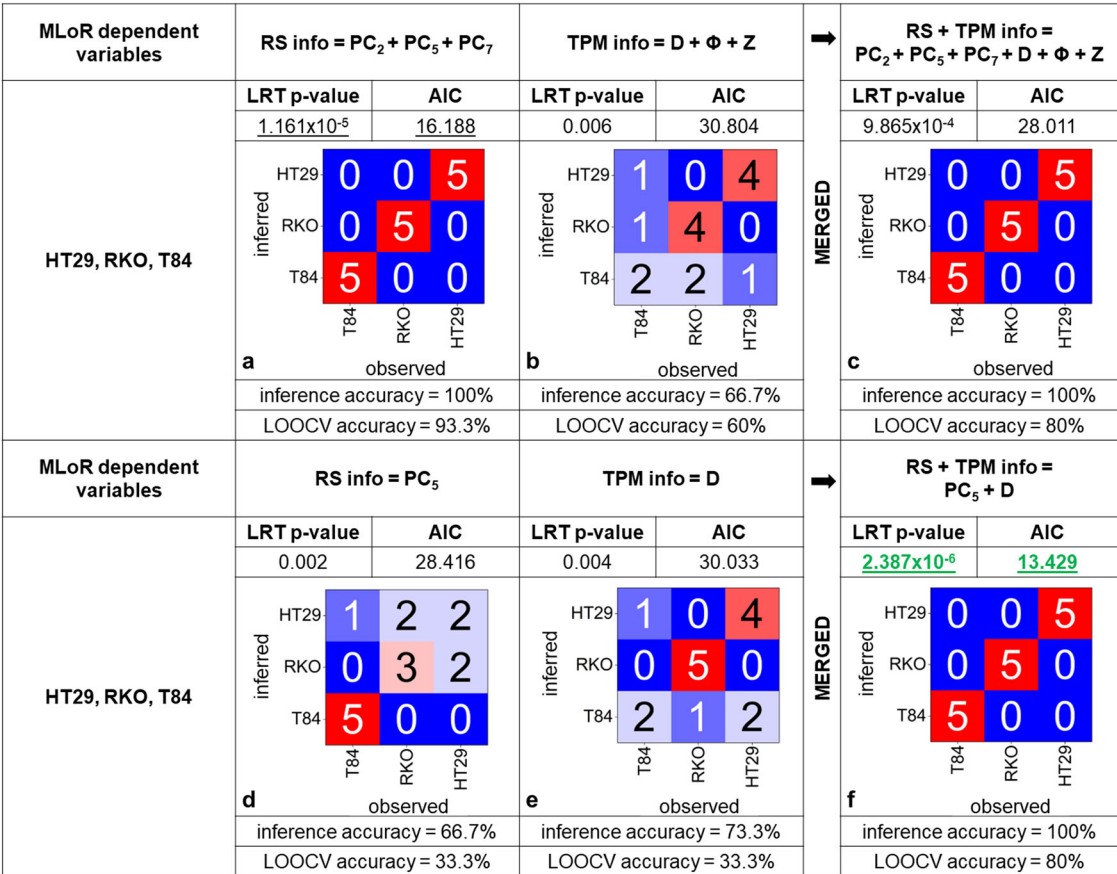

**Fig. 7 | MLoR models statistics to infer colon cancer types: the reduced RS + TPM model shows superior LRT and AIC metrics. a** MLoR model fed with all the statistically significant PCs extracted from the RS dataset analysis. **b** MLoR model using all the TPM-derived indexes showing significantly different trends across cell types. **c** Enlarged MLoR model using all the information fed in **a** and **b**. **d** Reduced MLoR model using only the most significant chemical information differentiating cell types, $PC_5$. **e** Reduced MLoR model using only the most significant morphological information differentiating cell types, D. **f** Best model in terms of LRT and AIC metrics, achieved coupling the one most significant chemical and morphological information, $PC_5$ and D. Inference and LOOCV accuracy are not significantly different from the ones obtained with the extended RS model (see also Supplementary Fig. 9).

model (Fig. 7d): reducing the number of PCs used as regressors has a negative impact on model performance, as expected, because PCs are orthogonal to each other. Conversely, exploiting only D among the TPM-derived information improves model performance (LRT *p*-value = 0.004, AIC = 30.033, and inference accuracy 73.3%) (Fig. 7e), which is explained by the high correlation among TPM observables (Fig. 6). The best model (in green in Fig. 7) is achieved when combining the reduced morpho-chemical information, $PC_5$ and D (Fig. 7f), through which we achieved superior discrimination of similar cancer phenotypes (LRT *p*-value = $2.387 \times 10^{-6}$, AIC = 13.429, and inference accuracy = 100%).

Overall, our phenotype inference analysis proves that using more explanatory variables does not lead to better model metrics, neither in the goodness of the fit per se (i.e., LRT) nor in the fit accuracy weighted by model complexity (i.e., AIC). Also, a higher number of independent explanatory variables jeopardizes model interpretability, as the effect of one predictor cannot be investigated separately, and dependence effects among variables may confound inference. The use of reduced morpho-molecular information increases inference accuracy (Fig. 7f) while easing interpretation to favor biological understanding.

As this work gives access to co-occurring morphological and chemical quantitative data, we investigate the relation between RS and TPM information in cells. It is acknowledged that the cell DM is mostly explained by the mass of protein content[13,35], which in turn produces predominant Raman peaks in the fingerpint region[34]. We tested experimentally and quantitatively such morpho-chemical proportionality. As the Raman peak intensity scales linearly with the concentration of scatterers in the focal

volume, we integrated the area under the curves of major protein-related Raman modes (i.e., phenylalanine at $\Omega = 1007$ cm$^{-1}$ and Amide I at $\Omega = 1660$ cm$^{-1}$) to obtain a semi-quantitative value representing the overall cellular protein content. We have access to the DM amount of these same cells via TPM. By linear fit of these morpho-chemical pairs (Fig. 8), we obtain a PCC = 0.977, with a significantly positive slope ($p = 6.85 \times 10^{-11}$), proving high positive linear correlation between cell protein content and cell DM. Nonetheless, major protein peaks cannot compensate the role of morphological information in phenotyping (Supplementary Fig. 8), at least when dealing with cell types not featuring DM as a significant distinguishing trait (Fig. 5b).

## Discussion

This work investigates the informative content, biological interpretability, and phenotyping potential of combined RS and TPM, a label-free non-invasive approach for living cell characterization. The data structure of RS and TPM information clearly differs. Here, we propose two systematic human-bias free pipelines to extract complementary molecular and morphological information from RS and TPM data for cell type differentiation, making sure to preserve biological interpretability. We broke down quantitatively the individual or synergistic contribution of native morpho-molecular traits in cell type inference. Notably, only when combining the most significant chemical and morphological explanatory variables (i.e., $PC_5$ and DM), one can obtain exquisite phenotype inference accuracy by means of a simplified MLoR mathematical model (Fig. 7). This clearly indicates that the combination of RS and TPM is not only valuable per se due

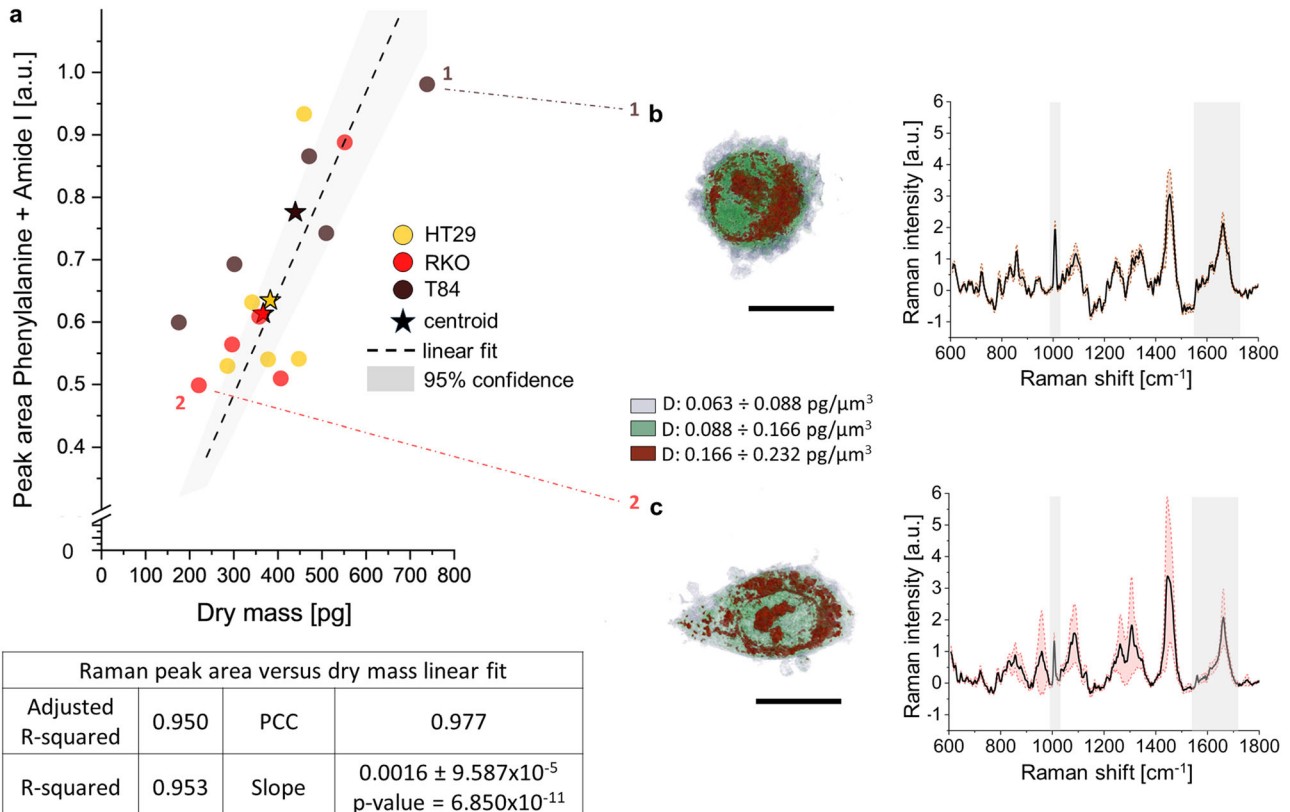

**Fig. 8 | RS-derived protein peaks integral is linearly related to TPM-derived protein dry mass. a** The scatter plot y-axis shows values of the area under the Raman peak curves of phenylalanine and Amide I, two predominant protein-related peaks. The x-axis reports values of the DM via TPM. The dashed line is the linear fit of the dataset with intercept fixed at the origin. Sample size = 5 independent cells/

phenotype. The table reports fitting statistics and slope value ± standard deviation. D distribution (exploited together with V to calculate the cell DM) and Raman peaks of cells corresponding to **b** point 1 (T84) and **c** point 2 (RKO) are shown. Scale bars: 10 μm.

to the higher and diverse amount of extracted information characterizing a single unperturbed cell, but this synergy is required to boost phenotyping accuracy: the model using the best RS explanatory variable (i.e., $PC_5$) achieves 66.7% inference accuracy with LRT $p$-value = 0.002 and AIC = 28.416; the model using the best TPM explanatory variable (i.e., D) achieves 73.3% inference accuracy with LRT $p$-value = 0.004 and AIC = 30.033, whereas the combined morpho-molecular model achieves 100% type inference accuracy with LRT $p$-value = $2.387 \times 10^{-6}$ and AIC = 13.429. This holds true even when tackling the discrimination of very similar cell types, such as the human colon cancer-derived cell subtypes in this work. Notably, our results exploited a limited sample size while accounting for biological variability and model overfitting (Supplementary Fig. 4 and Supplementary Fig. 5), ultimately saving time and costs. Investigating the predictive power of this morpho-chemical dataset on unseen examples is beyond the scope of this work. Still, we observe that the same reduced subset of morpho-molecular data is key to ensure high phenotype prediction accuracy as well, as shown through leave-one-out cross validation (LOOCV) (Supplementary Fig. 9). To further study prediction performance, enlarged datasets should be generated in future research.

Furthermore, this work investigates any shared information between RS and TPM maps, mainly given by the positive linear correlation (PCC = 0.977) of cell protein content[52], as encoded into the intensity of their molecular vibrations and into their refractive behavior[13,35,53], two clearly different observables. In fact, correlating RS and TPM-derived traits in proteins is important: TPM may be useful to monitor drug response, especially when the drug affects cellular proteins, such as proteasome inhibitors. This further confirms the soundness of FOV co-registration and theory alignment of our experiments.

Although our morpho-molecular cell-phenotyping strategy works with cell-averaged data that are easily and quickly extracted form RS and TPM maps, we demonstrate that this approach can also be easily applied for intra-cellular organelles characterization (Supplementary Fig.10), where RS proves to be a valid chemical probe to tell apart subcellular structures with comparable RI in TPM maps, overcoming their intrinsic lack of chemical specificity.

It is worth noticing that the acquisition time of an RS and a TPM map differ: while the former generally takes from tens of minutes to hours, the latter is much faster, generally taking a few seconds. Hence, we envision TPM as an effective pre-screening tool to speed up live-cell characterization, to be coupled to co-registered RS mapping at a second stage to achieve exquisite phenotyping accuracy and cell characterization[21]. Also, the speed mismatch between RS and TPM can be overcome by using coherent Raman scattering (CRS) techniques[8,54,55], which typically target one Raman shift at a time, selected based on the RS full-spectrum analysis (e.g., here, addressing only $PC_5$-specific peaks). CRS would reduce chemical imaging time, making it comparable with TPM.

To foster a wider transferability of our methods in biology and biotechnology, combining RS and TPM imaging into one single plug-and-play platform would be beneficial: it would limit the engineering skills required from the operator and ease the method by avoiding transportation and co-registration of the analytes across systems. Furthermore, combined RS and TPM have a potential for non-invasive monitoring of cancer progression in living organisms. The need for an epi-detection scheme and the limited accessibility of target cells in in-vivo scenarios are challenges, but recent reports confirm the effectiveness of in-vivo TPM[56,57] or RS[58,59], mainly exploiting oblique illumination schemes, fiber-optics and endoscopy.

The presented methods and tools, not requiring any sample manipulation with respect to standard culture conditions, ultimately stand as an effective, fast, accurate and practical solution for cell type characterization and discrimination in biomedical and cancer research, also in the common scenario of short supply of biological material thanks to the robustness of the approach to small sample sizes.

## Methods

### Cell culture

This study employs HT29 colorectal adenocarcinoma cells (ATCC, product code HT29 HTB-38, USA), RKO colon carcinoma cells (ATCC, product code RKO CRL-2577, USA), and T84 cells of colon cancer lung metastasis (ATCC, product code T84 CCl-248, USA). HT29s were cultured in base medium consisting of McCoy's 5a (Modified) Medium (ThermoFisher Scientific, catalog number: 16600082, USA), supplemented with fetal bovine serum (FBS) (Millipore Sigma, catalog number: F0926, USA) to a final concentration of 10%, and 5000 U/mL Penicillin-Streptomycin (PenStrep) (ThermoFisher Scientific, catalog number: 15070063, USA) to a final concentration of 1%. RKO cells were cultured in Eagle's Minimum Essential Medium as a base medium, adding 10% FBS and 1% PenStrep. T84 cells were cultured in Dulbecco's Modified Eagle's Medium (DMEM): F-12 Medium (ThermoFisher Scientific, catalog number: 21041025, USA) supplemented with 5% FBS and 1% PenStrep. Cells were expanded in 250 mL cell culture flasks (Avantor by VWR, catalog number: 10062-862, USA) and passaged every 48 hours using Trypsin/EDTA (Millipore Sigma, catalog number: T4049, USA) for detaching from substrates. Cells were transferred to 35 mm quartz-bottom Petri dishes (WakenBTech, catalog number: SF-S-D12, Japan) at least 12 hours before RS and TPM measurements, to ensure cell adhesion to the uncoated substrate. To ease the localization of designated target cells when switching from the RS to the TPM microscopy system, so to deliver co-registered chemical and morphological maps, quartz bottoms were marked with a reference grid. Details about our approach to ease a quick co-registration of RS and TPM maps of single living and label-free cells are presented in Supplementary Fig. 11 (see also Supplementary Fig. 10). Through culture and measurements, cell physiological conditions were granted.

### RS system

This study employs a home-built high-throughput microscope capable of multi-position and multi-timepoint point-scanning Raman microscopy. The laser source is a 785-nm continuous-wave Ti-Sapphire laser cavity (Spectra-Physics, model: 3900 S) with average output power of 750 mW, pumped by a 532-nm laser operating at 5-W average power (Spectra-Physics, model: Millennia eV). An inverted Olympus IX83 fluorescence microscope body is integrated with such 785-nm Raman excitation laser coupled to the backport, where a 749-nm dichroic mirror (ThorLabs, model: DMSP750B) deflects the excitation to the sample through an Olympus UPLSAPO $60 \times 1.2$ NA water-immersion objective. The average laser power at the backport of the microscope is tuned to be 100 mW via a polarizer, achieving 75 mW average laser power at the sample plane. Similarly, white light is coupled for bright field imaging. The bright field and Raman imaging modes are switched by swapping dichroic filters with auto-turrets. Galvo mirror-based scanning and stage scanning are combined to acquire each single FOV and multiple FOVs, respectively, delivering high-throughput Raman measurements. A custom MATLAB (2020b) script communicates with an open-source microscope control software (Micro-manager)[60], a digital acquisition (DAQ) board, and signal detection to automatize data acquisition. The script pipeline allows for a multi-dimensional measurement consisting of brightfield, fluorescence and Raman channels at multiple positions and z-stacks. The backscattered light is collimated by the same objective. Through the 785 nm dichroic mirror, visible light (brightfield and fluorescence signals) is short-pass filtered from the Raman scattering signal and sent to a NIR spectrograph with a 5.03 nm/mm nominal dispersion grating having central wavelength at 750 nm (Andor, Holospec HS-HSG-785-LF) and a detector CCD camera (Princeton Instruments, model: PIXIS

100BR eXcelon). The fluorescence and brightfield channels are imaged by the Orca Flash 4.0 v2 sCMOS camera from Hamamatsu Photonics. The exposure time for each point in the Raman measurement is 1.5 s. This implies a light dose of 261.63 kJ/cm² at the sample plane, which is significantly lower compared to previously reported values that ensured noninvasiveness in living cells[61]. Also, we systematically maintained cells in culture for at least two passages after measurements occurred, to experimentally confirm the noninvasiveness of our methods: neither cell proliferation abnormalities nor nonstandard cell detachment was observed. FOV is $50 \times 50$ pixels, $40 \times 40$ μm², with each pixel corresponding to $800 \times 800$ nm². The time to acquire Raman hyperspectral images is 1 hour per FOV. Evaporation of the immersion water for the objective is compensated by a home-built automated water-immersion feeder using syringe pumps and syringe needles glued to the tip of the objective lens. Here, water was supplied at a flow rate of 1 μL/min. To conduct live cell imaging, samples are incubated throughout measurement using a top stage incubator to maintain physiological conditions (Tokai Hit, model: stxg-welsx-set). It distributes heat uniformly within the incubation chamber through a top heater leaving the bottom free for optical access for the inverted configuration objective. An integrated temperature sensor allows real-time feedback regulation to achieve temperature control at 37 °C. The incubation set also keeps the humidity level inside the chamber > 95% by heating distilled water in an internal peripheral bath unit. An automated gas blender controller unit connected to a 100% $CO_2$ gas cylinder mixes $CO_2$ with surrounding air obtaining 5% $CO_2$ and 95% air as an input to the chamber. Wavenumber calibration is regularly achieved exploiting standard acetamidophenol samples to annotate known Raman peaks and the Raman shift axis is derived by quadratic interpolation. We systematically set the Z coordinate of the RS imaging plane by maximizing the signal counts on the CCD camera. We observed that such a position along the Z axis coincided with $+ 5$ μm with respect to the imaging plane not showing any trace of cell-related Raman peaks, due to a focal spot falling fairly inside the quartz substrate.

### RS data processing

Raman hyperspectral maps are post-processed via MATLAB (2023a) and "RamApp", a web-based tool developed internally[62], with the following ordered steps. Through "RamApp", spectra are corrected for cosmic rays employing the built-in spike removal tool. Then, the fluorescence baseline of each spectral pixel is fit according to the adaptive smoothness partial least squares algorithm, with a choice of $\lambda = 5 \times 10^6$ as the smoothness parameter[63], and subtracted. Spectral noise correction via Savitzky-Golay filtering (employing a second-order polynomial) is used as a spectral denoiser, with a 7-points filter window. Each Raman hyperspectral map is normalized by dividing it by its Frobenius norm. Through MATLAB (2021b), PCA is computed in the processed Raman maps, and $PC_1$-projected images clearly distinguish cell pixels (high pixel values in the projected image) from background ones (low pixel values in the projected Raman map). Hence, cell pixels selected by high $PC_1$ values are averaged into one representative spectrum per cell, and imported in Origin(Pro) (Version 2022. OriginLab Corporation, Northampton, MA, USA) for the further statistical analysis.

### TPM system

The quantitative phase images of cells were obtained with a commercial tomographic phase microscope (Tomocube HT-2H, Tomocube Inc., South Korea)[64]. It uses a $60 \times 1.2$ NA water-immersion objective that, coupled with a 532-nm green excitation laser, leads to lateral and axial resolutions of ~110 nm and ~356 nm, respectively. Theoretical calculation and experimental demonstration of such spatial resolution are reported in greater detail by Park and coworkers[65], using the same optical setup. Tomograms are reconstructed using the diffraction tomography algorithm with the Rytov approximation on a set of 100 2D phase images scanned in a circular pattern with a DMD-driven illumination angle of 49° with respect to the optical axis[64]. Each 2D phase image fed into the algorithm is obtained from

spatially modulated interferograms through the field retrieval algorithm[66]. All TPM maps in this study, covering a volume of $40 \times 40 \times 40\ \mu m^3$, corresponding to $425 \times 425 \times 210$ pixels, were taken encompassing the same corresponding FOV in Raman hyperspectral maps of cells. To conduct live-cell imaging, we exploited a dedicated commercial incubation chamber providing physiological conditions (Tomo Chamber, Tomocube Inc., South Korea).

## TPM data processing
TPM holographic tomograms of cancer cells were analyzed using TomoStudio software (Tomocube Inc., South Korea). The RI of PBS (RI = 1.3342) was used for background calibration. We applied 0.19 ml/g as the refraction increment, which is widely established by previous reports[36], and RI = 1.345 as the threshold to run a volumetric segmentation of the target cell, so as to restrict to these voxels the computation of TPM morphological indexes. Such extracted data are imported in Origin(Pro) for further statistical analysis.

## Statistics and reproducibility
For RS data analysis, PCA is run on the standardized dataset (mean = 0 and standard deviation = 1 to obtain a normal data distribution) through the PCA for Spectroscopy Origin(PRO) add-on. MLR is performed on PC scores and cell types through the Origin(Pro) MLR built-in function, and statistical significance of MLR β coefficients is calculated through a two-tailed $t$-test with a 0.05 significance level. For TPM data analysis, statistical significance of V, S, A, RI, D, T, DM, and Φ across HT29, RKO and T84 cell types is computed via U-Mann Whitney non-parametric test, with a significance level of 0.05, to allow for a non-normal distribution of data points. Significance of PCC values in correlation analyses is calculated via a two-tailed $t$-test, with a significance level of 0.05. MLoR models are fit on RS and TPM-derived data using the Logistic Regression Origin(Pro) add-on, which includes LRT and AIC built-in algorithms for model statistics and evaluation.

To sample biological variability, HT29, RKO and T84 cells included in this work are independently cultured in different plates: we produced $n = 5$ independent biological replicates per each of the three cancer cell subtypes.

## Reporting summary
Further information on research design is available in the Nature Portfolio Reporting Summary linked to this article.

## Data availability
All data needed to support the conclusions in this paper are present in the paper and in the Supplementary Materials and Supplementary Data. Source data and analyses output data can be found in the open-access Zenodo repository[67] titled Supplementary material - Superior Label-free Morpho-Molecular Phenotyping of Living Cancer Cells by Combined Raman Spectroscopy and Phase Tomography, at the following link: https://doi.org/10.5281/zenodo.10779109. Other data are available from the corresponding author on reasonable request.

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

## Acknowledgements

This research was funded by the European Union's Horizon 2020 research and innovation program under grant agreement No. 101016923 and by the European Innovation Council (EIC) under grant agreement No. 101047137 and No. 101058004, by NIH grants P41EB015871 and UG3CA275687, and by the Progetto Rocca of the MIT-Italy foundation. D.P. and G.C. acknowledge financial support by the European Union's NextGenerationEU Programme with the I-PHOQS Infrastructure [IR0000016, ID D2B8D520, CUP B53C22001750006] "Integrated infrastructure initiative in Photonic and Quantum Sciences". A.B. acknowledges the award of a Progetto Rocca Doctoral Fellowship to conduct collaborative research at Massachusetts Institute of Technology and Politecnico di Milano.

## Author contributions

Conceptualization, A.B. and J.W.K.; methodology, A.B., J.W.K., K.J.K.K.; software, A.B. and K.J.K.K.; formal analysis, A.B., J.W.K., R.V., experimental acquisition of Raman spectra and phase tomograms: A.B., cell culture: A.B., sample preparation: A.B., data processing and statistics: A.B., writing - original draft preparation, A.B., J.W.K.; writing - review and editing A.B.,

K.J.K.K., G.N.F.C., R.V., P.T.C.S., D.P., and J.W.K., visualization, A.B.; supervision, J.W.K., R.V., D.P., and P.T.C.S.; funding acquisition, A.B., D.P., and P.T.C.S. All authors have read and agreed to the published version of the manuscript.

## Competing interests

The authors declare no competing interests.
