## [Peer Review File · Communications Biology]

Reviewers' comments:

Reviewer #1 (Remarks to the Author):

The authors present an overall interesting study showing that Raman spectroscopy and tomographic phase microscopy can provide complementary information for improved cell classification. Overall I think this paper is well matched for the scope of Communications Biology and will be of interest to the readers. There are a few points where the authors should further clarify:

Major comment:

If some Raman peaks are correlated with DM (Fig. 8), can those be used instead of DM in Fig. 7(F)? How would that affect performance?

Minor comments:

How did the authors ensure the same cells were collected on the two different systems? Perhaps a little more discussion/clarity on the limitation of using two different systems to collect the data would be good.

Would looking at intra-cellular spatial correlations of TPM vs. RS signatures be possible? How closely can the signatures be spatially matched?

For RS, what is the laser power on the sample? The manuscript say two different things: "The laser power entering the microscopy unit is tuned to be 100 mW via a polarizer" and "Laser power at the sample plane is 120 mW." What is the damage threshold?

Reviewer #2 (Remarks to the Author):

The work presented in this manuscript, "Superior Label-free Morpho-Molecular Phenotyping of Living Cancer Cells by Combined Raman Spectroscopy and Phase Tomography", is highly relevant and valuable to the field of biomedical research, particularly in cancer cell phenotyping and disease progression monitoring. The authors have addressed a pressing need in the life sciences by developing a rapid, non-invasive and label-free approach to the characterisation of live cancer cells. The combination of RS and TPM provides a comprehensive morpho-molecular profile of the cells, allowing the study of independent or correlated chemical and morphological features.

Comments and questions to the authors:

1. In the Results section, the authors mentioned that five living cells were considered for each cell line to reduce the number of materials and measurement time while sampling biological variability. It would be helpful to provide additional justification or references supporting the sufficiency of this sample size for reliable statistical analysis and inference.
2. The authors mentioned the potential of their approach for non-invasive monitoring of cancer progression in living organisms. While this is an exciting prospect, it would be beneficial to discuss any limitations or challenges that may arise in translating this technique from cell culture studies to in vivo applications.
3. The authors mentioned in the context that "It uses a 60X 1.2 NA water immersion objective that, coupled with a 532 nm green excitation laser, leads to lateral and axial resolutions of ~ 110 nm and ~ 356 nm, respectively." As the lateral and axial resolutions are finer than regular point spread function estimation using a 532 nm laser and NA1.2 objective, can the authors provide the resolution measurements of this TPM system?
4. Given the sentences in the paper: "Cell cycles were not synchronised through the work, so to make results agnostic to sampled cell cycle phases." Can the authors provide evidence and comments on the effects of Raman and TPM data at different cell cycle phases?

Also, spontaneous Raman measurement of a single cell took more than one hour, did the authors find cell movement/proliferation after one hour? Can different imaging depths of Raman mapping of cells influence the results?

5. Please add more practical details or experimental images about "To ease the localisation of designated target cells when switching from the RS to the TPM microscopy system, so to deliver co-registered chemical and morphological maps, quartz bottoms were marked with a reference grid.", as it is an essential step to carry out effective morpho-molecular phenotyping.

Overall, this manuscript presents a novel and promising approach for label-free morpho-molecular phenotyping of living cancer cells. The work is well-executed, and the findings are significant for the field of biomedical research. With the recommended additions and clarifications, this manuscript would be a valuable contribution to the scientific literature. Based on the quality of the work and the potential impact, I would recommend the editors to accept this paper for publication after addressing the comments and questions raised above.

Cambridge, May 14th 2024

Manuscript ID: COMMSBIO-24-1099

We deeply thank the Reviewers for their careful consideration of our manuscript entitled “*Superior Label-free Morpho-Molecular Phenotyping of Living Cancer Cells by Combined Raman Spectroscopy and Phase Tomography*”. We appreciate the Reviewers for their recommendation to publish our work pending revisions.

In the following, we addressed in detail the Reviewers’ concerns and discuss the related edits to the manuscript. Thanks to the Reviewers’ guidance, we substantially expanded the data analysis presented in our manuscript, which we consider beneficial to ensure the readers’ comprehensive understanding. Also, we presented in greater detail our experimental procedures of morpho-molecular co-registration, which will benefit the reproducibility of our results.

We (i) shortened the text of our manuscript by slight rephrasing, making it compliant with the recommended lengths for Abstract, Introduction, Results and Discussion sections; (ii) produced corrected Figures, generating box plots instead of bar plots (Fig. 5) and corrected minor Figure typos (Fig. 7); (iii) included all experimental data, analysis output data, and graphs source data used through the work in an open-access Zenodo repository, linked in the Data Availability statement.

Please note the following: Text in **blue font** corresponds to the Reviewers' comments, in **black font** is our response to the Reviewers' comments, and **text highlighted in yellow** represents the changes made in the revised manuscript.

Reviewer #1

The authors present an overall interesting study showing that Raman spectroscopy and tomographic phase microscopy can provide complementary information for improved cell classification. Overall I think this paper is well matched for the scope of Communications Biology and will be of interest to the readers. There are a few points where the authors should further clarify.

We truly appreciate the Reviewer's positive feedback on our manuscript. We thank the Reviewer for the insightful notes and guidance to improve our work: they strengthened the rigor of the presented data analysis and results, ultimately increasing the readers' comprehension.

1. Major comment: If some Raman peaks are correlated with DM (Fig. 8), can those be used instead of DM in Fig. 7(F)? How would that affect performance?

We thank the Reviewer for the insightful comment. In fact, major protein-related Raman peaks positively correlate with TPM-derived cell dry mass (DM) (Fig. 8), but it is the TPM-derived average cell density (D), not the overall cell DM, that acts as the most significant morphological trait to effectively distinguish similar cancer cell types in living and label-free conditions (Fig. 7F). Despite this, we followed the Reviewer's suggestion and investigated the role of such protein-related Raman peak areas in replacing the information about cell D (Fig. 7F) as an independent variable for phenotype inference. Although RS peak areas show positive correlation with TPM-derived cell DM (PCC=0.977), they cannot outperform the high phenotype inference accuracy obtained with the direct use of TPM-derived cell D in the reduced morpho-molecular model. We attribute this result to two main reasons: (i) RS protein-related peak areas, computed on cell-averaged Raman spectra, correlate significantly with the cell DM, not with the cell average D that is the one TPM-derived trait mostly distinguishing colon cancer cell types (Fig. 7). In fact, Figure 5B shows that the cells DM does not vary significantly among these very similar cancer cell types. Using an RS-based regressor that scales with the cells DM cannot benefit cell type inference; (ii) even if the cells DM were a significant TPM-based information distinguishing cancer cell types, replacing it with a positively correlated RS-based independent variable in a regression model would not achieve equal performance. In fact, phenylalanine and Amide I are the major protein-related Raman peaks in the fingerprint region of the Raman spectrum, but cannot comprehensively recapitulate the plethora of cell proteins which contribute to the cell DM. Hence, we consider that having access to the overall cell DM via TPM is a powerful information, hardly achievable with equal quantitative rigor via the integration of major RS peaks found in the fingerprint region of Raman spectra.

ACTION TAKEN:

1. In *Supplementary Information, Figure S8 and Lines 105 – 125*, we presented the comparison of performances between MLoR models using Raman peak areas or TPM-derived density (D), in terms of LRT p-value, AIC, inference accuracy and LOOCV prediction accuracy:

MLoR dependent variables	RS info = PC ₅		RS protein peak areas		→	RS info = PC ₅ + RS protein peak areas	
	LRT p-value	AIC	LRT p-value	AIC		LRT p-value	AIC
HT-29, RKO, T-84	0.002	28.416	0.218	37.913	MERGED	0.012	32.104
	inferred		inferred			inferred	
	(A)	observed	(B)	observed		(C)	observed
	inference accuracy = 66.7%		inference accuracy = 53.3%			inference accuracy = 60%	
	LOOCV accuracy = 33.3%		LOOCV accuracy = 13.3%		LOOCV accuracy = 20%		
MLoR dependent variables	RS info = PC ₅		TPM info = D		→	RS + TPM info = PC ₅ + D	
	LRT p-value	AIC	LRT p-value	AIC		LRT p-value	AIC
HT-29, RKO, T-84	0.002	28.416	0.004	30.033	MERGED	2.387x10⁻⁶	13.429
	inferred		inferred			inferred	
	(D)	observed	(E)	observed		(F)	observed
	inference accuracy = 66.7%		inference accuracy = 73.3%			inference accuracy = 100%	
	LOOCV accuracy = 33.3%		LOOCV accuracy = 33.3%		LOOCV accuracy = 80%		

Figure S8. Comparison of MLoR models performances using protein-related RS peaks instead of TPM-derived D to infer colon cancer types. Despite the positive correlation that RS peak areas show with TPM-derived cell DM, they cannot outperform the high phenotype inference accuracy obtained with the use of TPM-derived cell D in the reduced morpho-molecular model. In fact, protein-related RS peak areas, computed on cell-averaged Raman spectra, correlate significantly with the overall cell DM, not with the cell average D that is the one TPM-derived trait mostly distinguishing colon cancer cell types (Fig. 7F). The cell DM does not vary significantly among the very similar cancer cell types used through this work (Fig. 5B). Using an RS-based regressor that scales with the cells DM cannot benefit cell type inference, as quantified statistically in the figure. (A, D) Reduced MLoR model using only the most significant chemical information differentiating cell types, PC₅. (B) Reduced MLoR model using only protein-related RS peaks of phenylalanine at $\Omega = 1007 \text{ cm}^{-1}$ and Amide I at $\Omega = 1660 \text{ cm}^{-1}$ as independent variables, which were proved to have a significantly positive correlation (PCC = 0.977) with the cells DM (Fig. 8). (C) Enlarged RS-based MLoR model using all the information fed in A and B. This model uses only the chemical information, with the RS peak areas. (E) Reduced MLoR model using only the most significant morphological information differentiating cell types, D. (F) Best model in terms of LRT and AIC metrics, achieved coupling the one most significant chemical and morphological information, PC₅ and D. Despite the correlation between RS peak areas and the cells DM, related to the cells D via their V (*i.e.*, $DM = D \times V$), the RS-based reduced information (C) cannot achieve equally high phenotype inference accuracy compared to the combined RS and TPM reduced model (F).

2. In Results, Cell type discrimination via combined morpho-molecular profiling, Lines 365 – 367, we referenced the supplementary material in the above:

“Nonetheless, major protein peaks cannot compensate the role of morphological information in phenotyping (Fig. S8), at least when dealing with cell types not featuring DM as a significant distinguishing trait (Fig. 5B).”

Minor comments:

2. How did the authors ensure the same cells were collected on the two different systems? Perhaps a little more discussion/clarity on the limitation of using two different systems to collect the data would be good.

We appreciate the Reviewer’s concern as this point needs better clarification in our manuscript to ensure the reproducibility of our results and the actual application of the presented methods and tools. We chose to expand our original statement in the Methods section with Supplementary Information, providing a graphical representation of the grid design on the quartz bottom of culture dishes, together with pictures of the resulting culture plate and a detailed description of the co-registration procedure. This method proved a simple yet extremely effective solution to quickly find the same target cell while switching from the RS system to the TPM system. Furthermore, we agree that the limitations of not having one single RS and TPM multimodal platform are not described sufficiently, despite certainly being a major outlook of the presented methods and tools. Using the same system would speed up the overall measurement time and decrease the experimental complexity, as no transportation of the analytes across systems nor any search for the same field of view would be necessary. As a consequence, having a single plug-and-play platform would certainly favor a wider transferability of our phenotyping approach to purely biological and biotechnological research scenarios, limiting the engineering skills required for operation. We presented these limitations and the consequent research direction in the conclusion of our manuscript.

ACTION TAKEN:

1. In *Supplementary Information, Figure S9 and Lines 126 – 150*, we thoroughly detailed the co-registration procedure exploiting the reference grid method:

“Figure S9. Grid design on quartz bottom culture dishes for quick co-registration of single-cell FOVs between the RS and TPM systems. Ahead of cell culture, we carefully drew with the help of a ruler and a permanent ink pen the illustrated grid on the external side of quartz bottom disks of culture dishes, along with an indication of quadrant numbers. Then, once the dish was introduced into the incubation chamber of the RS system, we designated a target FOV among all the available grid angles (indicated with red dashed circles in the above sketch). The criteria for such a choice were: (i) having one single cell close to the grid angle, with a clear separation space between surrounding cells; (ii) all

the cell parts were adhering onto a portion of the quartz substrate not featuring any black ink traces, which would induce light scattering and alter the collected signal. After RS acquisition, the dish was transferred into the incubation chamber of the adjacent TPM system. Here, the same grid angle was quickly identified with the help of quadrant numbers, and the previously imaged single cell could be localized next to the edges of the grid angle. With this, the target cell was quickly and easily identified, enabling co-registered TPM measurements. Due to the short transfer time from the RS system to the TPM system, we did not experience cell division starting and terminating through transfer time. From time to time, major cell movement or a neat cell division occurred through RS imaging, due to its longer acquisition time (*i.e.*, circa 1 hour) compared to systems switch or TPM imaging, which caused strong blurring in the chemical map. For the sake of producing a clear morpho-molecular co-registration of RS and TPM images, carrying out a valid proof of concept study on the effectiveness of morpho-molecular phenotyping, such blurred RS images were excluded from the study. Conversely, minor cell reshaping and minimal cell movement was expected, as living cells were maintained in their physiological culture conditions inside incubation chambers. This behavior is totally acceptable and does not alter the soundness of the co-registration between the cell-averaged Raman spectrum and the cell-averaged TPM-derived traits. Notably, it supports the non-invasiveness of the methods. Such a minimal displacement can be clearly seen in Figure S5, where the two brightfield imaging modalities in the RS and TPM systems were used to take a picture of the designated single cell.”

2. In *Methods, Cell Culture, Lines 448 – 449*, we referenced the Supplementary Information in the above:

“Details about our approach to ease a quick co-registration of RS and TPM maps of single living and label-free cells are presented in Figure S9.”

3. In *Discussion, Lines 418 – 421*, we addressed the limitations of having separated systems for chemical and morphological imaging, pinpointing the advantages of a single multimodal platform, which stands as a major outlook for this study:

“To foster a wider transferability of our methods in biology and biotechnology, combining RS and TPM imaging into one single plug-and-play platform would be beneficial: it would limit the engineering skills required from the operator and ease the method by avoiding transportation and co-registration of the analytes across systems.”

3. Would looking at intra-cellular spatial correlations of TPM vs. RS signatures be possible? How closely can the signatures be spatially matched?

We thank the Reviewer for this comment. This point was not discussed in our manuscript, yet it would be of great interest to the readership. Indeed, the major drawback of TPM is its lack of chemical specificity, while it can reconstruct subcellular structures and morphology with exquisite spatial resolution (lateral and axial resolutions of ≈ 110 nm and ≈ 356 nm, respectively). Notably, this intra-cellular investigation is not needed for the sake of cell phenotyping, as we demonstrated that the cell-averaged information, easier and quicker to be extracted, can effectively and significantly distinguish similar cell types in label-free and living conditions. Also, due to expected slight cell motion occurring over the measurement time, as living cell conditions were ensured, an exact spatial overlap of RS and TPM maps is unlikely, whereas the cell-averaged morpho-molecular properties used for cell phenotyping through our study are robust to such minor movements. Nevertheless, showing that a reasonable intra-cellular spatial correlation is present would be beneficial for the audience seeking to use RS as a chemical probe into subcellular structures present in phase tomograms. A pixel-wise co-registration would not be rigorous, due to the intrinsic differences that RS and TPM imaging feature,

which ultimately lead to different pixel/voxel dimension. Still, as we co-registered FOVs with the same dimension and similar cell orientation, we confirm that it is possible to segment subcellular structures based on their RS fingerprint, then apply the same mask to retrieve their location in a central XY slice of the RI tomogram. RS fingerprints effectively distinguish intra-cellular structures based on their chemical composition, allowing for the differentiation of organelles that feature the same refractive index (Fig. S10). The analysis in the following proves this point and clarifies the amount of spatial correlation one should expect subcellular organelles to have through the presented bimodal approach.

ACTION TAKEN:

1. In *Supplementary Information, Figure S10 and Lines 153 – 179*, we showed the spatial correspondence between RS and TPM intra-cellular structures one should expect considering minor cell movements through measurements, highlighting the possibility of using our multimodal method to chemically differentiate structures with comparable RI values:

“Figure S10. Co-registered RS and TPM allow for the chemical differentiation of intra-cellular organelles with similar refractive properties. Thanks to the co-registration of FOVs with equal dimension and comparable cell orientation, we prove that it is possible to segment subcellular structures based on the intensity of their RS fingerprint signal, then apply the same mask to retrieve their location in a central XY slice of the corresponding RI tomogram. With this, one can achieve the differentiation of intra-cellular structures revealed with high spatial resolution (lateral and axial resolutions of ≈ 110 nm and ≈ 356 nm, respectively) in TPM maps that feature the same RI. Notably, this intra-cellular investigation is not necessary for the sake of cell phenotyping, as we proved that the cell-averaged information, easier and quicker to be extracted, can effectively and significantly distinguish similar cell types in label-free and living conditions (Fig. 7). Also, due to an minor cell motion that we expect

occurring over measurement time, as living cell conditions were ensured, an exact spatial overlap of RS and TPM maps is unlikely, whereas the cell-averaged morpho-molecular properties used for cell phenotyping through our study are robust to such slight movements. Scale bars are 10 μm . Signal thresholding and mask generation were carried out in Fiji-ImageJ. Signal threshold was automated through the IsoData function. (A) Morpho-molecular profile of a living and label-free human colon adenoma HT29 cancer cell in culture, likely undergoing cell division as suggested from its bi-nucleated structure. (B) Segmentation of cytoplasmic predominant portions of the target cell, through intensity thresholding of the Raman image at $\Omega = 715 - 725 \text{ cm}^{-1}$, corresponding to the Raman modes of $\text{CN}^+(\text{CH}_3)_3$ in lipids, choline group $\text{N}^+(\text{CH}_3)_3$ and phosphatidylcholine vibrations. The mask is applied to the central XY slice of the TPM map of the same cell, showing high spatial correspondence achieved despite minor cell motion. (C) Segmentation of the lipid accumulations in the cytoplasm of the HT29 cell, by intensity thresholding of the Raman hyperspectral map in the range $\Omega = 1440 - 1450 \text{ cm}^{-1}$, featuring the bending of CH_2 bonds in fatty acids. The lipid-related mask is applied to the central XY slice of the corresponding phase tomogram: despite the presence of similar RI structures throughout the cell, only a subset of them can be identified as cytoplasmic lipid accumulations thanks to co-registered chemical mapping. (D) Segmentation of the nucleic acids-rich region of the cell via intensity thresholding of the Raman hypercube at $\Omega = 780 - 790 \text{ cm}^{-1}$, featuring pyrimidines ring breathing, including both nucleobases of DNA and RNA cytosine and uracyl at $\Omega = 788 \text{ cm}^{-1}$. The mask is applied to the central XY slice of the TPM map, showing evident spatial correspondence with the higher RI structures inside the cell nuclei, suggesting their identification with nucleoli.”

2. In *Discussion*, Lines 405 – 409, we highlighted the prospective usage of our methods to differentiate intra-cellular organelles in TPM maps via RS:

“Although our morpho-molecular cell-phenotyping strategy works with cell-averaged data that are easily and quickly extracted from RS and TPM maps, in Figure S10 we demonstrate that this approach can also be easily applied for intra-cellular organelles characterization, where RS proves to be a valid chemical probe to tell apart subcellular structures with comparable RI in TPM maps, overcoming their intrinsic lack of chemical specificity.”

4. For RS, what is the laser power on the sample? The manuscript say two different things: “The laser power entering the microscopy unit is tuned to be 100 mW via a polarizer” and “Laser power at the sample plane is 120 mW.” What is the damage threshold?

We thank the Reviewer for pointing out this mistake in our original manuscript. We confirm that the laser power at the backport of the RS microscope amounted to 100 mW, corresponding to 75 mW at the focal plane after the objective. We corrected this mistake in the new version of the manuscript.

Regarding laser-induced photodamage thresholds in living cells, near-infrared (NIR) CW lasers delivering comparable light doses have been widely used in biomedical optics due to their lower energy photons with respect to visible light, which strongly benefits noninvasiveness [W. Katagiri *et al.*, *J. Biomed. Opt.*, vol. 25(3), pp. 036003, 2020; S. Waldchen *et al.*, *Sci. Rep.*, vol. 5, pp. 15348, 2015; Wagner, M., *Int. J. Mol. Sci.* vol. 11, pp. 956-966, 2010.]. A seminal work by Waldchen and coworkers⁵⁴ clearly showed that the energy of photons used to irradiate cells has a dramatic impact on phototoxicity: exposing label-free cultured cells to 488-nm CW laser irradiation for 240 s at a light dose of 44.85 kJ/cm^2 determined a cell survival rate of 0%, whereas 100% of the cells survived with a slight red shift to a 514-nm CW laser irradiation at a light dose of 46.38 kJ/cm^2 . At 640 nm, the light dose could reach 1414.61 kJ/cm^2 without causing cell photodamage (namely, 98% of survival rate). In our study, we employed a near IR CW laser source operating at 785 nm for RS imaging, substantially red-shifted with respect to the wavelengths employed by Waldchen and colleagues. Considering the 1.2 NA of our

objective, we operated with a light dose of 261.63 kJ/cm² having a pixel exposure time of 1.5 s. Hence, our illumination parameters, featuring markedly red-shifted photons and lower irradiation time, determine a significantly lower light dose (261.63 kJ/cm²) compared to previously reported values that ensured cell noninvasiveness (1414.61 kJ/cm²). Throughout this work, we systematically maintained cells in culture for at least two passages after measurements occurred, to experimentally confirm the noninvasiveness of our methods by monitoring cell proliferation over time. Neither cell proliferation abnormalities nor nonstandard cell detachment was observed.

ACTION TAKEN:

1. In *Methods, RS system, Lines 460 – 462*, we corrected the value of the average laser powers used in the RS system:

“The average laser power at the backport of the microscope is tuned to be 100 mW via a polarizer, achieving 75 mW average laser power at the sample plane.”

2. In *Methods, RS System, Lines 475 – 480*, we provided the readers with details regarding the non-invasive light dose we employed, and referenced earlier reports assessing the photodamage threshold of living cells under CW laser exposure:

“This implies a light dose of 261.63 kJ/cm² at the sample plane, which is significantly lower compared to previously reported values that ensured noninvasiveness in living cells⁵⁴. Also, we systematically maintained cells in culture for at least two passages after measurements occurred, to experimentally confirm the noninvasiveness of our methods: neither cell proliferation abnormalities nor nonstandard cell detachment was observed.”

3. In *Bibliography, Lines 677 – 678*, we added the following reference:

[60] Wäldchen, S., Lehmann, J., Klein, T., Van De Linde, S. & Sauer, M. Light-induced cell damage in live-cell super-resolution microscopy. *Sci Rep* **5**, 15348 (2015).

Reviewer #2

The work presented in this manuscript, "Superior Label-free Morpho-Molecular Phenotyping of Living Cancer Cells by Combined Raman Spectroscopy and Phase Tomography", is highly relevant and valuable to the field of biomedical research, particularly in cancer cell phenotyping and disease progression monitoring. The authors have addressed a pressing need in the life sciences by developing a rapid, non-invasive and label-free approach to the characterisation of live cancer cells. The combination of RS and TPM provides a comprehensive morpho-molecular profile of the cells, allowing the study of independent or correlated chemical and morphological features.

We sincerely thank the Reviewer for the positive feedback. Thanks to the valuable comments, we substantially extended the presented data analysis, which allowed us to strengthen the robustness of our results. We hope that the revised version of the manuscript will now allow the readers to achieve a more comprehensive understanding of our methodologies.

Comments and questions to the authors:

1. In the Results section, the authors mentioned that five living cells were considered for each cell line to reduce the number of materials and measurement time while sampling biological variability. It would

be helpful to provide additional justification or references supporting the sufficiency of this sample size for reliable statistical analysis and inference.

We thank the Reviewer for raising this concern. We agree that quantitatively justifying the reduced sample size with which our phenotyping method operates is of key importance to support the claim of validity of our tools in common scenarios of scarcity of biological materials. In addition to the statistical tests showing significant differences among cell phenotypes in the first version of our manuscript, we produced new statistical evidence, exploiting the Kruskal-Wallis analysis of variance (KW-ANOVA) test^{26,27}. This is a non-parametric conservative adaptation of the commonly used one-way ANOVA test, valid also when the assumptions of normality and homogeneity of variances are violated.

We applied the KW-ANOVA test to the morpho-molecular observables used to train extended and reduced MLoR inference and prediction models, selected based on their descriptive statistics: PC₂, PC₅ and PC₇ for the RS-based chemical information, and D, Φ , and Z, for the TPM-based morphological information. In the KW-ANOVA, if the p-value is less than the significance level (0.05), one can reject the null hypothesis and conclude that there are significant differences among at least some of the group means. To graphically visualize the statistical differences achieved with our chosen sample size (*i.e.*, five living cancer cells per phenotype), we produced mean rank paired comparisons plots (Fig. S11) for each morphological or molecular trait analyzed. Thanks to such plots, one can easily visualize which groups are driving the statistical difference in the KW-ANOVA test, if any: the higher the mean rank difference between two phenotypes, the higher the contribution of that cell type pair in driving the overall significance of the KW-ANOVA test. In addition, we carried out a Dunn's test on the pairwise comparisons: this *post-hoc* non-parametric test compares all possible pairs of groups using pairwise Mann-Whitney U tests, then corrects the p-values for multiple comparisons via a Bonferroni correction. This test is highly conservative, it strongly limits the false-positive discovery rate of statistical significance and may generate false negatives. Despite this, and with a limited sample size, our analysis shows that two observables in both the RS-based and TPM-based datasets feature clear statistically significant differences among cell phenotypes. We conclude that the KW-ANOVA and Dunn's tests confirm the sufficiency of our sample size for phenotyping. Due to the high amount of quantitative information that multimodal RS and TPM can extract from each single living cell, these tools are suited for phenotyping tasks even when the biological material is in short supply.

ACTION TAKEN:

1. In *Supplementary Materials, Figure S11 and Lines 185 – 207*, we presented the additional KW-ANOVA test combined with the Dunn's test, to quantify the presence of statistically significant differences showing in our reduced sample size:

“Figure S11. The non-parametric Kruskal-Wallis Analysis Of Variance (KW-ANOVA) test of the morpho-molecular observables mostly distinguishing colon cancer phenotypes confirms the sufficiency of a reduced sample size. We applied the KW-ANOVA test to the morpho-molecular observables used to train extended and reduced MLor inference and prediction models, selected based on their descriptive statistics: **(A) PC₂**, **(B) PC₅** and **(C) PC₇** for the RS-based chemical information, and **(D) D**, **(E) Φ**, and **(F) Z**, for the TPM-based morphological information. In the KW-ANOVA test, if the p-value is less than the significance level (0.05), one can reject the null hypothesis and conclude that there are significant differences among at least some of the group means. The KW-ANOVA test exact p-values are: **(A) p = 0.088**, **(B) p = 0.02**, **(C) p = 0.05**, **(D) p = 0.018**, **(E) p = 0.063**, **(F) p = 0.022**. To graphically visualize the statistical differences achieved with our chosen sample size (*i.e.*, five living cancer cells per phenotype), we produced mean rank paired comparisons plots for each morphological or molecular trait analyzed. Thanks to such plots, one can easily visualize which groups are driving the statistical difference in the KW-ANOVA test, if any: the higher the mean rank difference between two phenotypes, the higher the contribution of that cell type pair in driving the overall significance of the KW-ANOVA test. In addition, we carried out a Dunn’s test on the pairwise comparisons: this post-hoc non-parametric test compares all possible pairs of groups using pairwise Mann-Whitney U tests, then corrects the p-values for multiple comparisons via a Bonferroni correction. This test is highly conservative, it strongly limits the false-positive discovery rate of statistical significance and may generate false negatives. Despite this, and with a limited sample size, our analysis shows that two observables in both the RS-based and TPM-based datasets feature clear statistically significant differences among cell phenotypes. We conclude that the KW-ANOVA and Dunn’s tests confirm the sufficiency of our sample size for phenotyping. Due to the high amount of quantitative information that multimodal RS and TPM can extract from each single living cell, these tools are suited for phenotyping tasks even when the biological material is in short supply.”

2. In *Results, Quantitative morpho-chemical cell mapping via co-registered RS and TPM, Lines 110 – 112*, we referenced the reader to the KW-ANOVA and Dunn’s tests, to support the sufficiency of the chosen sample size:

“For each human colon cancer cell line (*i.e.*, HT29, RKO, and T84), five living cells are considered (see also Fig. S1, S2, and S3), in order to reduce the materials and the measurement time while sampling biological variability^{26,27} (see Fig. S7 and Fig. S11).”

3. In *Supplementary Information, Figure S7, Lines 99 – 102*, we added reference to the confirmatory evidence of sufficiency of our limited sample size, as quantified above:

“Further confirmatory evidence of the sufficiency of the sample size used through this work can be found in Figure S11, which directly studies the intra and inter-phenotype variance of the morpho-molecular variables used to train the full and reduced MLoR inference models.”

4. In *Bibliography, Lines 598 – 601*, we added the following references to the theory of the statistical tests used for the additional analysis in the reviewed manuscript:

[26] Hecke, T. V. Power study of anova versus Kruskal-Wallis test. *Journal of Statistics and Management Systems* **15**, 241–247 (2012).

[27] Dinno, A. Nonparametric Pairwise Multiple Comparisons in Independent Groups using Dunn’s Test. *The Stata Journal* **15**, 292–300 (2015).

2. The authors mentioned the potential of their approach for non-invasive monitoring of cancer progression in living organisms. While this is an exciting prospect, it would be beneficial to discuss any limitations or challenges that may arise in translating this technique from cell culture studies to *in vivo* applications.

We thank the Reviewer for suggesting a clearer explanation of the prospective outlook of translating our methods for *in-vivo* applications, such as *in-vivo* cancer progression monitoring and cell phenotyping. We recognize this prospect was not detailed in the first version of this work. In our revision, we commented on recent literature reporting the effectiveness of using TPM^{55,56} and RS⁵⁷ in an epi-detection mode in *in-vivo* skin, and efforts are underway to integrate such microscopy modalities in endoscopy platforms⁵⁸. Hence, we are confident that ongoing technological developments will foster the *in-vivo* translation of superior non-invasive morpho-molecular phenotyping, here shown as an effective strategy even when limiting population sampling.

ACTION TAKEN:

1. In *Discussion, Lines 421 – 425*, we discussed the challenges of a prospective *in-vivo* application of RS and TPM for cell phenotyping and monitoring of cancer progression:

“Furthermore, combined RS and TPM have a potential for non-invasive monitoring of cancer progression in living organisms. Despite the need for an epi-detection scheme and the limited accessibility of target cells in *in-vivo* scenarios, recent reports confirm the effectiveness of *in-vivo* TPM^{55,56} or RS^{57,58}, mainly exploiting oblique illumination schemes, fiber-optics and endoscopy.”

2. In *Bibliography, Lines 667 – 674*, we added references to recent seminal works showing the use of RS and TPM in *in-vivo* applications:

[55] Costa, P. C. *et al.* Towards *in-vivo* label-free detection of brain tumor margins with epi-illumination tomographic quantitative phase imaging. *Biomed. Opt. Express* **12**, 1621 (2021).

[56] Abraham, T. M. *et al.* Label- and slide-free tissue histology using 3D epi-mode quantitative phase imaging and virtual hematoxylin and eosin staining. *Optica* **10**, 1605 (2023).

[57] Kang, J. W. *et al.* Direct observation of glucose fingerprint using *in vivo* Raman spectroscopy. *Sci. Adv.* **6**, eaay5206 (2020).

[58] Lombardini, A. *et al.* High-resolution multimodal flexible coherent Raman endoscope. *Light Sci Appl* **7**, 10 (2018).

3. The authors mentioned in the context that "It uses a 60X 1.2 NA water immersion objective that, coupled with a 532 nm green excitation laser, leads to lateral and axial resolutions of ~ 110 nm and ~ 356 nm, respectively." As the lateral and axial resolutions are finer than regular point spread function estimation using a 532 nm laser and NA1.2 objective, can the authors provide the resolution measurements of this TPM system?

We thank the Reviewer for pointing out that further clarification about the spatial resolution of the TPM system is needed to explain the reason why it outperforms common resolution values achieved under standard illumination. The theory behind the calculation of the spatial resolution of the same TPM system here used is described in great detail by Park *et al.* in a separate publication⁶³. Briefly, the 3D resolution of TPM was systematically quantified and analyzed through numerical simulations using its optical transfer function (OTF), whose spatial bandwidth is the inverse of the Abbe limit, in the illumination scanning method here used. To demonstrate the feasibility of the simulated OTF in an experimental situation, they compared the OTF-derived theoretical point spread function (PSF) and the experimentally measured PSF of the TPM system (Fig. R1). Experimental data were collected with a 5 μ m-diameter silica bead. The experimental results with a silica bead correspond to the theoretically calculated resolution, achieving 110 nm and 350 nm for the lateral and axial direction, respectively. The slight experimental variation in axial resolution is due to noise and optical aberrations.

Figure R1. Experimental assessment of the TPM spatial resolution. (a) Cross-sectional slices of the reconstructed RI map of a silica bead in the XY, XZ, and YZ planes. (b) Phantom of an ideal 5- μ m-diameter silica bead. (c) Experimental PSF of the optical system acquired using the deconvolution of the experimentally measured tomogram and the phantom. (d) Experimental and theoretical PSF viewed at various polar angles. Adapted from Park C. and colleagues.

ACTION TAKEN:

1. In *Methods*, TPM system, Lines 520 – 521, we referenced the work described above to clarify to the reader the spatial resolution of the TPM system:

“Theoretical calculation and experimental demonstration of such spatial resolution are reported in greater detail by Park and coworkers⁵⁷, using the same optical setup.”

2. In *Bibliography, Lines 684 – 686*, we added the following reference:

[63] Park, C., Shin, S. & Park, Y. Generalized quantification of three-dimensional resolution in optical diffraction tomography using the projection of maximal spatial bandwidths. *J. Opt. Soc. Am. A* **35**, 1891 (2018).

4. Given the sentences in the paper: "Cell cycles were not synchronised through the work, so to make results agnostic to sampled cell cycle phases." Can the authors provide evidence and comments on the effects of Raman and TPM data at different cell cycle phases?

We appreciate the Reviewer's suggestion. We provided further supporting explanation regarding the effect of cell cycle phases on RS and TPM observables, and our methods being robust to this in the task of cell phenotyping. More specifically, the main phenotypic differences we could quantify in cells at neatly different cell cycle phases, namely, in interphase or undergoing mitosis, are related to dimensional TPM traits. These mainly include volume (V) [μm^3], surface area (S) [μm^2], projected area or footprint (A) [μm^2], total dry mass DM [pg], and, in minor proportion, cell thickness (T) [μm]. In fact, it is reasonable to observe enlarged cells (*i.e.*, higher values of V, S, A, and T) through the mitotic phase, when two nuclei are generated in the cytoplasm, and cells synthesize all the necessary proteins to ensure a complete set of organelles for two daughter cells (*i.e.*, an increased value of total DM, which scales with the content of dry proteins). We provide the data points of TPM traits of mono- and bi-nucleated colon cancer cells included in our dataset (Fig. S12), from which such trends can be clearly evinced.

Conversely, when observing the PC scores of RS spectra of cells displaying one or two nuclei, no appreciable difference is present. This is perfectly in line with the fact that cells undergoing final interphase and progressing through mitosis produce a double set of cytoplasmic compounds and double their nuclei. Hence, we expect the cell-averaged RS spectra of mono- or bi-nucleated cells of the same phenotype to be similar to each other, as they feature the same average biomolecular composition. This is proven by the comparable values of the first 10 PC scores, representing 98.9 % of the dataset variance, of the RS data points of mono- and bi-nucleated cells of the same phenotype included in our study (Fig. S13).

ACTION TAKEN:

1. In *Supplementary Information, Figure S 12 and Lines 210 - 221*, we present the values of the complete set of eight TPM-derived observables as they vary among cell types at interphase or towards mitosis, namely, when displaying one or two nuclei in their cytoplasm:

Figure S12. Impact of different cell cycle phases on TPM observables. We did not synchronize cell cycles to make our methods agnostic to them and easily and directly applicable to pristine unmanipulated cells in culture. When cells are in a binucleated state, over mitosis, the main phenotypic differences are given by an enlarged cell dimension (*i.e.*, **(A)** surface area (S) [μm^2], **(B)** volume (V) [μm^3], **(C)** projected area or footprint (A) [μm^2], and, in minor proportion, **(D)** cell thickness (T) [μm]), along with an increase in average cell dry mass (*i.e.*, **(E)** DM [pg]). **(F-H)** Other observables do not show appreciable variations among cell cycle phases. In fact, it is reasonable to observe enlarged cells through the mitotic phase, when two nuclei are generated in the cytoplasm and cells synthesize all the necessary proteins to ensure a complete set of organelles for two daughter cells (namely, an increased value of total DM, which scales with the content of dry proteins present). We provide the data points of TPM traits of mono- and bi-nucleated colon cancer cells belonging to the same cell type, included in our dataset, from which such trends can be clearly evinced.

2. In *Supplementary Information, Figure S 13, Lines 224 – 237*, we present the values of the first ten PC scores of the RS dataset, as they vary among cell types at interphase or towards mitosis, namely, when displaying one or two nuclei in their cytoplasm:

“Figure S13. Impact of different cell cycle phases on RS observables. We did not synchronize cell cycles to make our methods agnostic to them and easily and directly applicable to pristine unmanipulated cells in culture. While cells vary their TPM-derived dimensional and mass traits from interphase (generally mononucleated) to mitosis (insurgence of binucleated cells) (Fig. S12), RS fingerprints do not show signs of clear variations. Due to cells doubling their nucleus and synthesizing proteins to provide a complete set of cytoplasmic compounds and organelles to two daughter cells, the overall chemical composition of a doubling cell, as represented by the cell-averaged Raman spectrum, should be reasonably similar to the chemical composition of the mononucleated mother cell. (A – L) We provide data points of the first ten PCs of mononucleated versus binucleated cells, belonging to the same phenotype, as present in our dataset, representing 98.9 % of cumulative variance of the RS data cloud. It is evident that the PC scores of binucleated cells do not clearly set apart from PC scores of mononucleated cells, in each PC considered: the overall chemical composition probed by RS is generally invariant to the sampled cell cycle phase. These results demonstrate the robustness of our method in being agnostic to cell cycles for the phenotyping task.”

3. In *Introduction, Lines 115 – 116*, we referred to the Supplementary Materials in the above:

“Cell cycles were not synchronized through the work, so as to make results agnostic to sampled cell cycle phases (see Fig. S12 and Fig. S13).”

5. Also, spontaneous Raman measurement of a single cell took more than one hour, did the authors find cell movement/proliferation after one hour? Can different imaging depths of Raman mapping of cells influence the results?

ANSWER:

We recognize this interesting point was not addressed in our initial submission; we have introduced this information in the revised manuscript. Indeed, we expected slight cell motion to occur over the measurements time, as living cell conditions were ensured. This behavior is totally acceptable and does not alter the soundness of the co-registration between the cell-averaged Raman spectrum and the cell-averaged TPM-derived traits used for phenotype inference. Due to the short transfer time from the RS system to the TPM system, we did not experience cell division starting and terminating through transfer time. Conversely, from time to time, major cell movement or a neat cell division occurred through RS imaging, due to its longer acquisition time (*i.e.*, circa 1 hour) compared to systems switch or TPM imaging, which caused strong blurring in the chemical map. For the sake of producing a clear morpho-molecular co-registration of RS and TPM images, to carry out a valid proof-of-concept study on the effectiveness of morpho-molecular phenotyping, such blurred RS images were excluded from the study.

As for the imaging depth of RS mapping, choosing a focal plane close to the cell borders causes a decrease of the signal-to-noise ratio of Raman peaks, either overwhelmed by quartz signals (if focusing towards the cell lower side), or overwhelmed by medium signals (if focusing towards the upper side of the cell). Hence, we systematically chose the Z coordinate by maximizing the Raman signal counts on the CCD camera. We observed that such a position along the Z axis coincided with + 5 μm with respect to the imaging plane not showing any trace of cell-related Raman peaks, due to a focal spot falling completely inside the quartz substrate. For the sake of reproducibility of our results and to foster the correct implementation of our methods, these experimental details were included in the revised manuscript.

ACTION TAKEN:

1. In *Supplementary Information, Figure S9 and Lines 139 - 150*, we clarified the presence of expected minor cell movements between RS and TPM mapping (please, refer to the reply to Reviewer #1, Comment #2, for extended details):

“Due to the short transfer time from the RS system to the TPM system, we did not experience cell division starting and terminating through transfer time. From time to time, major cell movement or a neat cell division occurred through RS imaging, due to its longer acquisition time (*i.e.*, circa 1 hour) compared to systems switch or TPM imaging, which caused strong blurring in the chemical map. For the sake of producing a clear morpho-molecular co-registration of RS and TPM images, to carry out a valid proof-of-concept study on the effectiveness of morpho-molecular phenotyping, such blurred RS images were excluded from the study. Conversely, minor cell reshaping and minimal cell movement were expected, as living cells were maintained in their physiological culture conditions inside incubation chambers. This behavior is totally acceptable and does not alter the soundness of the co-registration between the cell-averaged Raman spectrum and the cell-averaged TPM-derived traits. Notably, it supports the non-invasiveness of the methods. Such a minimal displacement can be clearly seen in Figure S5 and Figure S10.”

2. In *Supplementary Information, Figure S10, Lines 161 – 164*, we clarified that minor cell motion was expected and did not compromise the co-registration of cell-averaged features, also allowing for a visual co-registration of intra-cellular organelles to chemically differentiate TPM structures (please, refer to the reply to Reviewer #1, Point #3, for extended details):

“Also, due to an expected minor cell motion occurring over measurement time, as living cell conditions were ensured, an exact spatial overlap of RS and TPM maps is unlikely, whereas the cell-averaged

morpho-molecular properties used for cell phenotyping through our study are robust to such slight movements.”

3. In *Methods, RS system, Lines 493 – 497*, we added details regarding the Z position of the imaging plane in RS measurements:

“We systematically set the Z coordinate of the RS imaging plane by maximizing the signal counts on the CCD camera. We observed that such a position along the Z axis coincided with + 5 μm with respect to the imaging plane not showing any trace of cell-related Raman peaks, due to a focal spot falling fairly inside the quartz substrate.”

6. Please add more practical details or experimental images about "To ease the localisation of designated target cells when switching from the RS to the TPM microscopy system, so to deliver co-registered chemical and morphological maps, quartz bottoms were marked with a reference grid.", as it is an essential step to carry out effective morpho-molecular phenotyping.

ANSWER:

We agree that the colocalization procedure was not comprehensively detailed in our manuscript and we thank the Reviewer for raising this concern. We included a schematical representation of the method along with a picture of a quartz-bottom culture dish marked with such a reference grid and detailed protocol explanation. As this concern perfectly aligns with Point #2 by Reviewer #1, we kindly ask to refer to the answer to this question for extended details.

ACTION TAKEN:

1. In *Supplementary Information, Figure S9 and Lines 126 – 150*, we thoroughly detailed the co-registration procedure exploiting the reference grid method (refer to the answer to Point #2 by Reviewer #1 to see the changes in the revised manuscript).
2. In *Methods, Cell Culture, Lines 448 – 449*, we referenced the Supplementary Information in the above.

Overall, this manuscript presents a novel and promising approach for label-free morpho-molecular phenotyping of living cancer cells. The work is well-executed, and the findings are significant for the field of biomedical research. With the recommended additions and clarifications, this manuscript would be a valuable contribution to the scientific literature. Based on the quality of the work and the potential impact, I would recommend the editors to accept this paper for publication after addressing the comments and questions raised above.

We thank the Reviewer for the very favorable judgement expressed on our manuscript and for recommending its publication. In our revised version, we have fully addressed the comments and questions from the Reviewer.

REVIEWERS' COMMENTS:

Reviewer #1 (Remarks to the Author):

The authors provided a thoughtful and thorough response to reviewer comments, and I recommend the paper is appropriate for publication.

Reviewer #2 (Remarks to the Author):

Dear Authors,

I have carefully reviewed your revised manuscript and the accompanying rebuttal letter. I appreciate the detailed responses you have provided to address the comments and questions raised by myself and the other reviewer.

You have successfully addressed all the concerns raised in the first review. The additional analyses and supplementary materials provided have significantly strengthened the manuscript. Furthermore, the discussion of the potential challenges and limitations in translating your approach to in vivo applications and the additional details on the co-registration procedure have improved the clarity and reproducibility of your work.

I am satisfied with the revisions, and I believe that the manuscript has been substantially improved. The work presented is well-executed, and the findings are significant for the field of biomedical research, particularly in cancer cell phenotyping and disease progression monitoring.

Based on the quality of the work and the potential impact, I would recommend that the editors accept this paper for publication in Communications Biology. Thank you for your efforts in addressing the reviewers' comments and questions. I wish you success with the publication of your research.